# Synaptically-targeted long non-coding RNA SLAMR promotes structural plasticity by increasing translation and CaMKII activity

Isabel Espadas [1,4], Jenna L. Wingfield[1,4], Yoshihisa Nakahata [2], Kaushik Chanda [1], Eddie Grinman[1], Ilika Ghosh [2], Karl E. Bauer[3], Bindu Raveendra[1], Michael A. Kiebler [3], Ryohei Yasuda [2], Vidhya Rangaraju [2] & Sathyanarayanan Puthanveettil [1] ✉

Long noncoding RNAs (lncRNAs) play crucial roles in maintaining cell homeostasis and function. However, it remains largely unknown whether and how neuronal activity impacts the transcriptional regulation of lncRNAs, or if this leads to synapse-related changes and contributes to the formation of long-term memories. Here, we report the identification of a lncRNA, SLAMR, which becomes enriched in CA1-hippocampal neurons upon contextual fear conditioning but not in CA3 neurons. SLAMR is transported along dendrites via the molecular motor KIF5C and is recruited to the synapse upon stimulation. Loss of function of SLAMR reduces dendritic complexity and impairs activity-dependent changes in spine structural plasticity and translation. Gain of function of SLAMR, in contrast, enhances dendritic complexity, spine density, and translation. Analyses of the SLAMR interactome reveal its association with CaMKIIα protein through a 220-nucleotide element also involved in SLAMR transport. A CaMKII reporter reveals a basal reduction in CaMKII activity with SLAMR loss-of-function. Furthermore, the selective loss of SLAMR function in CA1 disrupts the consolidation of fear memory in male mice, without affecting their acquisition, recall, or extinction, or spatial memory. Together, these results provide new molecular and functional insight into activity-dependent changes at the synapse and consolidation of contextual fear.

Specific alterations in transcription[1–3], localized translation[4,5], and axonal transport[6–9] lead to the creation of new synapses and the modification of existing ones[10,11]. These are widely recognized processes responsible for the establishment of long-term memory (LTM). Nevertheless, the precise mechanisms governing these critical steps and their spatial-temporal regulation remain poorly understood.

Transcriptional changes associated with learning are very intricate. They rely on multiple components of the transcriptome undergoing unique changes in specific neuronal populations for LTM.

Recent advancements in next-generation sequencing technology have unveiled the complexity of the transcriptome and led to the discovery of novel noncoding RNA families. Among these, long noncoding RNAs (lncRNAs) are particularly intriguing due to their roles in driving epigenetic changes in the nucleus and regulating translation in the cytoplasm, hinting at their potential as pivotal mediators of LTM[12–14]. Notably, approximately 40% of these lncRNAs are specifically enriched in the brain[15,16]. Recent studies, including our own, have suggested that lncRNAs may play a critical role in fundamental neuronal functions,

[1]Department of Neuroscience, The Herbert Wertheim UF Scripps Institute for Biomedical Innovation & Technology, Jupiter, FL, USA. [2]Max Planck Florida Institute for Neuroscience, Jupiter, FL, USA. [3]Biomedical Center, Department for Cell Biology, Ludwig-Maximilians-University of Munich, Medical Faculty, 82152 Planegg-Martinsried, Germany. [4]These authors contributed equally: Isabel Espadas, Jenna L. Wingfield. ✉e-mail: sputhanveettil@ufl.edu

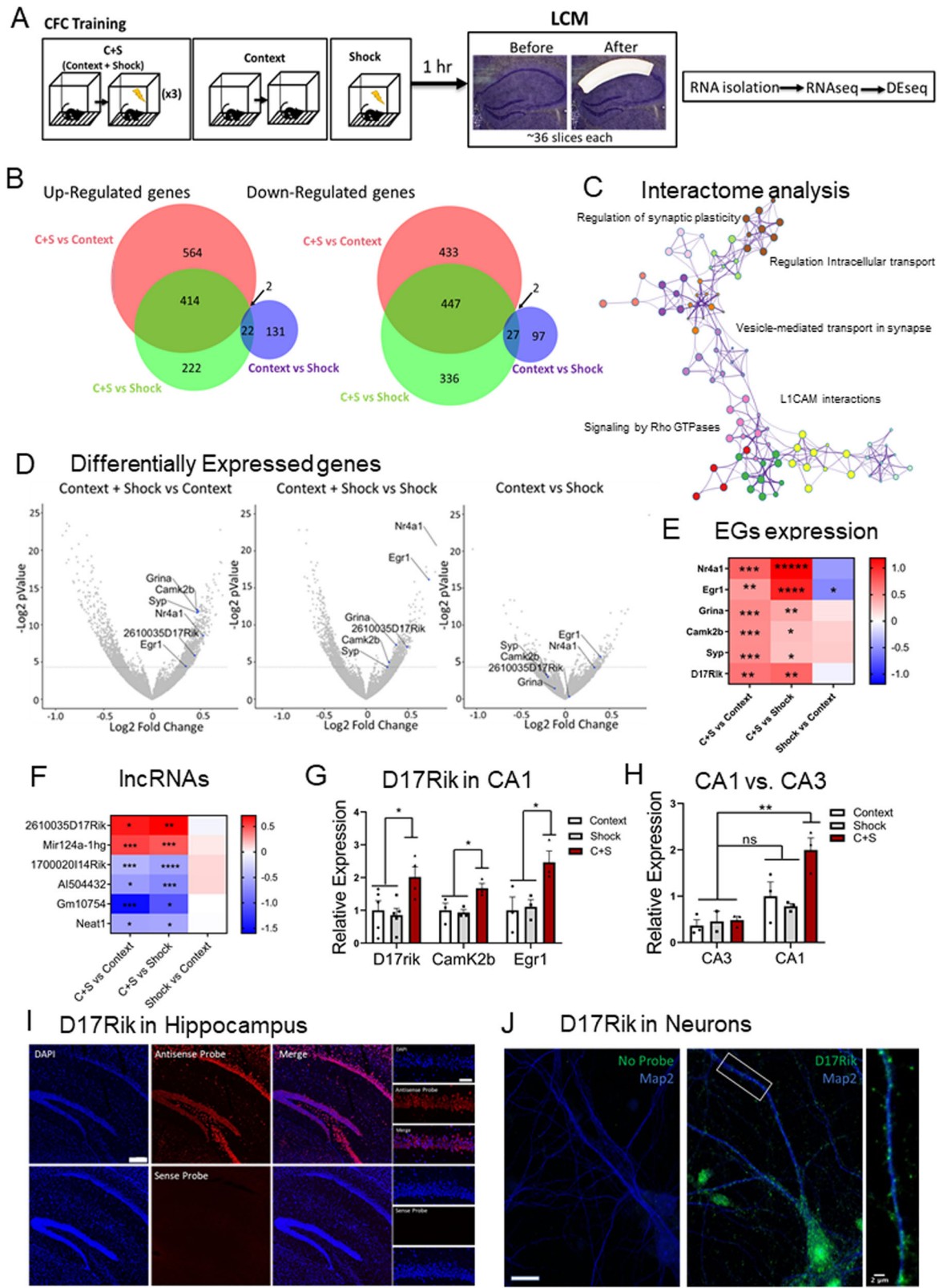

such as synaptic remodeling, transmission, synaptogenesis, neurogenesis, and neuronal differentiation[12–14,17–23]. Moreover, previous research has shown that different brain regions implicated in memory, such as the prefrontal cortex (PFC), amygdala, and hippocampus, exhibit unique lncRNA profiles[24,25]. Specifically, the hippocampus displays distinct lncRNA expression patterns in different subregions of the tri-synaptic circuitry[25]. These specific patterns suggest that

lncRNAs may have distinct roles in modulating neuronal functions during LTM. Additionally, changes in lncRNA expression patterns can be regulated in an activity-dependent manner[12,13,23,26–29].

While our understanding of the neurobiology of long noncoding RNAs (lncRNAs) has made significant progress, their specific functions in mediating subcellular signaling and influencing neuronal plasticity remain largely unclear. In pursuit of this question, we explored

**Fig. 1 | lncRNA D17Rik is enriched in dorsal CA1 of mice after CFC.**
**A** Experimental Design. 1 hr after CFC training brains were fast frozen. The tissue was stained with RNAse-free cresyl violet. The dorsal CA1 was dissected by LCM, processed for RNA isolation followed by RNAseq and DEseq in "R". **B** Venn diagrams derived from the DEseq analysis show a higher number of significantly regulated genes in the experimental group (C + S) compared to control groups (context alone and immediate shock) (nominal p-value < 0.05). **C** Metascape analysis indicates that the majority of genes significantly regulated in C + S condition compared to controls are grouped into a cluster related to early synaptic plasticity. **D** DEseq results represented by volcano plots show a higher enrichment of the lncRNA D17Rik and plasticity-related genes in the C + S group compared to controls (nominal p-value < 0.05). **E** Heat map represents log2Foldchange of EGs differentially expression in CFC. **F** Heat map represents log2 Foldchange of some of the most differentially regulated lncRNAs by C + S in dorsal CA1. **G** Relative expression of D17rik (Context $n = 5$, Shock $n = 5$, C + S n=4 mouse hippocampi) and synaptic plasticity-related genes CamK2b and Egr1 ($n = 3$ for all conditions) 1 h after training (One-Way ANOVA, Multiple Comparisons Dunnett's test, D17rik *$p = 0.03$, CamK2b *$p = 0.03$, Egr1 *$p = 0.04$; data is shown as mean ± SEM). **H** D17Rik is enriched only in CA1 after CFC in the C + S, but not in CA3. There are no significant differences between CA1 and CA3 D17Rik expression in control groups ($n = 3$-4). Two-way ANOVA + Tukey's test. **$p = 0.002$; data is shown as mean ± SEM. **I** FISH shows that the lncRNA D17Rik is expressed in mouse hippocampus. High magnification details from a representative mouse's pyramidal layer in CA1 indicate a mainly cytoplasmic subcellular localization of D17Rik. Photomicrographs show D17Rik(red) and nuclear marker DAPI(blue). Scale bars = 200 μm. This experiment was completed with tissues from 3 mice. Each time produced similar results. **J** FISH photomicrograph shows that D17Rik is expressed in mouse primary hippocampal neuronal cultures. D17Rik (green) colocalizes with dendritic marker Map2(blue). Cell body scale bar=20 μm. Dendrite inset scale bar=2 μm. This experiment was completed four different times. Each time produced similar results.

whether contextual fear conditioning (CFC), a type of associative learning known for inducing robust and enduring fear memories, could trigger the expression of lncRNAs in distinct neuronal populations and whether they played varying roles in different forms of LTM.

We, therefore, conducted unbiased analyses of gene expression in CA1-hippocampal neurons to identify lncRNA changes induced by CFC. This led to the discovery of a lncRNA we termed SLAMR (Synaptically Localized Activity Modulated lncRNA), which exhibited enrichment in the dorsal CA1 region. SLAMR is primarily found in the cytosol and is transported to neuronal dendrites and into the spine compartment. Through time-lapse quantitative imaging and 2-photon glutamate uncaging to stimulate individual spines, we observed that SLAMR is trafficked within dendrites via KIF5C, a molecular motor protein, and controls dendritic complexity, translation, and activity-dependent changes in synaptic structure. Furthermore, we identified a critical sequence element in SLAMR that is essential for its proper transport and interaction with specific proteins. Notably, we observed that changes in SLAMR expression had a significant impact on the activity of CaMKIIα in synaptoneurosomes, with a loss of SLAMR function resulting in reduced CaMKII activity in CA1 pyramidal neurons. Additionally, restricting SLAMR expression in the CA1 region impaired the consolidation of CFC, while spatial memory remained unaffected. These findings collectively demonstrate that the lncRNA SLAMR plays a pivotal role in hippocampal-dependent associative long-term memory, influencing various molecular, structural, and functional processes.

## Results
### CFC induces specific transcriptional changes in the coding and noncoding transcriptome of the dorsal CA1

In search of transcriptional changes in the CA1 modulated by experience in mice, we used CFC training, a behavioral paradigm that establishes robust LTM storage and requires the participation of this hippocampal subarea[30]. This behavioral paradigm provides the advantage of having distinct protocols for independently analyzing each memory phase (i.e., acquisition, consolidation, extinction, and recall). To specifically identify lncRNAs differentially regulated in the early phases of memory, 1 hour after CFC, we isolated the dorsal CA1 by laser capture microdissection (LCM) and extracted RNAs for total RNAseq from a total of 6 mice divided into the three different groups. LCM isolated CA1 RNAs from shock-alone (S) or context-alone (C) groups were used as controls for identifying differentially expressed genes (DEGs) induced by experience (Fig. 1A). The Venn diagrams shown in Fig. 1B indicate the total number of genes in the transcriptome, that were significantly up-regulated or down-regulated based on a nominal p-value < 0.05 (~400 DEGs), compared with the two control groups, context-alone, and shock-alone (GSE214838) (Supplementary Data S1B). In contrast, the comparison between control conditions identified only ~100 DEGs significantly different

between groups (p-value < 0.05). These results suggest that CFC induced significant and specific transcriptional changes in dorsal-CA1 neurons.

The CFC-induced DEGs (p-value < 0.05) largely encompassed genes involved in synaptic transmission and synaptic plasticity (Fig. 1C). Interactome analysis using Metascape revealed that the enriched pathways are mainly related within a cluster encompassing the regulation of synaptic plasticity, vesicle transport at the synapse, L1CAM interactions, and Rho GTPases signaling (p-adjusted <0.05, Fig. 1C). We next searched for the presence of well-known mRNAs related to plasticity processes induced by neuronal activity to confirm the efficiency of the CFC training. Specifically, we characterized the unique transcriptome profile in the dorsal CA1 for CFC training. The analysis of the genes that were upregulated in the C + S compared to both individual control groups, revealed several genes related to short- and long-term plasticity processes. Cellular component analysis indicated a significant enrichment in several synapse communication pathways like postsynaptic density, postsynaptic specialization, and neuron-to-neuron synapses (p-adjusted<0.0005; Supplementary Fig. S1A and Table S1A). Biological process analysis indicated a large enrichment in neuron specialization and synaptic plasticity, protein modification, and localization, among others (p-adjusted<0.05, Supplementary Fig. S1B and Table S1B) suggesting an increase in different metabolic processes and plasticity changes in CA1 neurons induced by experience. Furthermore, the molecular function analysis showed regulation of transcriptional and translational processes induced by CFC in the dorsal CA1 with significant changes in genes related to mitochondrial activity, specifically, genes involved in NADH activity (ex. *Ndufa10, Ndufa11, Cbr1*) and ribonucleotide binding (ex. *Ckmt1, Matk, Tubb5*) (p-adjusted<0.05, Supplementary Fig. S1C, and Table S1C). Together, these data reveal noteworthy changes in the activity of CA1 dorsal neurons specifically induced by CFC compared to either shock-alone or context-alone controls.

Additionally, DEseq analysis of RNAseq data (Fig. 1D and E, Supplementary Data S1D, and E) identified critical genes exclusively upregulated in CFC-trained mice compared to controls. Some of these genes are also up-regulated in previous studies in the hippocampus of mice following CFC training or involved in hippocampal LTP: for example, *Egr-1*[31,32], the glutamate receptor *Grina, Camk2β*[33,34], *Nr4a1*[35,36] and *SYP* (Synaptophysin)[37]. Taken together, these results identified selective experience-dependent changes in the coding transcriptome of the CA1.

We next closely examined the noncoding RNAs identified by our DEseq analysis. In total, we identified 11 lncRNAs differentially expressed (DE) in the C + S condition compared to both control groups (Fig. 1F, Supplementary Data S1F). As lncRNAs are not easily identified in total RNAseq analysis from small and specific portions of the brain, these lncRNAs were selected based on a nominal p-value < 0.05 from the overlapped DEGs of both comparisons (C + S vs. Context, C + S vs.

Shock alone) with consistent results. Most of these lncRNAs have unknown functions, except for Neat1, a well-studied lncRNA that plays important roles in memory functions and stress responses in mice[28,38]. Neat1 is significantly downregulated in the C + S condition compared to context and shock alone ($p$-value < 0.05). As previous studies demonstrated, Neat1 is known to suppress the immediate-early gene c-Fos, whose activation is essential in the early stages of memory acquisition[28]. Unfortunately, studies following a similar methodology to ours are scarce, making comparative analysis difficult. Nevertheless, several of the undescribed lncRNAs found to be significantly regulated after CFC in our RNAseq have also been found to be significantly changed with CFC in global hippocampal lysates in other studies[39].

The resulting DE lncRNAs were then classified into different sub-categories based on their biotypes using the Ensemble annotation library (NCBIM37): sense_intronic, antisense, and long intergenic noncoding RNAs (lncRNAs), among others. Here, we identified 6 lincRNAs that showed significant changes in the C + S condition exclusively, indicating dynamic regulation related to hippocampal activity induced by CFC training in the dorsal-CA1 ($p$-value < 0.05, Fig. 1F). DEseq results showed that two of these genes (*2610035D17Rik*, *Mir124a-1hg*) were up-regulated ($p$-value < 0.05) in the C + S condition within the dorsal CA1. Both lncRNAs are strong candidates for being crucial regulators of neuronal plasticity processes associated with memory. In fact, Mir124a-1hg rat homolog, neuroLNC, has been implicated in neurogenesis and presynaptic activity[23].

## The lncRNA D17Rik is modulated by CFC in CA1 neurons

Among the upregulated lncRNAs in the CA1 of mice, we decided to focus on a previously undescribed lncRNA- 2610035D17Rik (D17Rik) as its expression is modulated in the CA1 by CFC. D17Rik is located in chromosome 11 and consists of 3 exons with non-coding potential indicated by phyloCSF and CPAT analysis[40] (Supplementary Fig. S1D, E). Independent validation by qRT-PCR analysis confirmed that this gene is significantly up-regulated only in the C + S group and its upregulation is also accompanied by an increase in the expression of two plasticity-related mRNAs, *CaMK2β* and *Egr-1* in the same samples ($n$ = 3-5 per group. One-Way ANOVA, Dunnett's test, Fig. 1G, Supplementary Data S1G), supporting the RNAseq findings. Interestingly, D17Rik expression in CA3 is not significantly enriched in any of the experimental groups (Fig. 1H, Supplementary Data S1H), suggesting region-specific regulation of D17Rik expression in the hippocampus. Together, these results demonstrate that D17Rik is an experience-dependent lncRNA specifically enriched in the dorsal CA1 hippocampal area following CFC training, suggesting a role in mediating contextual fear memory in the CA1.

We next examined the evolutionary conservation of D17Rik by searching for potential orthologs. Two possible D17Rik orthologs have been previously described in the literature in humans and zebrafish, the LINC00673 and LOC110366352 (SlincR), respectively[41–43]. The tissue specificity of SlincR in zebrafish is not known. The zebrafish transcriptome sequencing project (BioProject PRJEB1986) indicates that this lncRNA is preferentially expressed in the head of adult males. Similarly, LINC00673 in humans is well expressed in the brain according to the Illumina bodyMap2 transcriptome BioProject (PRJEB2445) and HPA RNA-seq normal tissues BioProject (PRJEB4337) in the NCBI databases. Interestingly, the loci for these two previously described theoretical orthologs are also conserved, especially regarding their position to other neighbor transcripts like *Sox9* and *Slc39a11*. In addition, this locus is also conserved in rats, where some lncRNAs show similar potential to bind *Sox9* promoter as well as conserved positions related to *Cog1*, *Slc39a11*, and *Sstr2*, particularly the lncRNA LOC102549836, a lncRNA with unknown functions highly expressed in the brain compared to other organs (BioProject PRJNA238328). The previously mentioned studies[42,43] suggested potential regulation of *Sox9* mRNA by these lncRNAs due to the

proximity (within 200 kb) of their promoters. As several lncRNAs are known to regulate other transcripts in a *cis*-manner, we decided to examine whether *SOX9* could be a target of D17Rik. Our RNAseq data showed a significant reduction of *Sox9* in the dorsal CA1 after CFC training exclusively in the C + S condition compared to context-alone (log2FC = −0.679; $p$-value = 0.0055) and shock-alone (log2FC = −0.677; $p$-value = 0.0054) control groups (Supplementary Fig. S1F). The log2-foldchange data for D17Rik and *Sox9*, indicates a negative correlation between them, suggesting a repression of the transcriptional processes (Supplementary Fig. S1G). To further examine this regulation, we carried out RNAi-mediated knockdown of D17Rik using antisense locked nucleic acid, Gapmer oligonucleotides in in vivo and in vitro models and assessed *Sox9* expression. Gapmers have previously been successfully used to knock down lncRNAs[13,22]. D17Rik silencing in our in vitro studies using mouse primary hippocampal cell cultures did not result in significant changes in the expression of the genes in the D17Rik locus, including *Sox9* (Supplementary Fig. S1H). Likewise, in our in vivo studies silencing D17Rik in the dorsal CA1 of mice hippocampus also showed no changes in *Sox9* when compared to the negative control. However, silencing D17Rik in vivo did show a significant reduction in the expression of the somatostatin receptor 2 (*Sstr2*) ($n$ = 7 each, *$p$ < 0.05, Unpaired t-test; Supplementary Fig. S1I). Together these data suggest that *SOX9* is not a direct target of D17Rik, but *Sstr2* may be.

To gain insight into the function of D17Rik, we next assessed its hippocampal and subcellular localization by fluorescence in situ hybridization (FISH) and qRT-PCR analysis of cellular fractions. FISH imaging showed almost no signal in the control sense probe condition, while the antisense probe images showed robust expression in different hippocampal subareas of the mouse brain (Fig. 1I, Supplementary Fig. S1J). FISH labeling in primary hippocampal neuronal cultures of mice revealed no labeling in any of the cellular compartments with the no-probe control, while the D17Rik-probe showed intense labeling in the cytoplasm of pyramidal neurons (Fig. 1J). To confirm this result, we analyzed cytoplasmic and nuclear fractions from mice hippocampi and analyzed D17Rik's expression by qRT-PCR. Using Actin as a cytoplasmic control for normalization and reference, we found that D17Rik is primarily expressed in the cytoplasm of the hippocampal neurons compared to the nucleus ($n$ = 3, *$p$ < 0.05, Student's t-test), while other well-known genes are equally distributed in both compartments (Map2), or specifically enriched in the nucleus ($n$ = 3 for both groups, Xist: *$p$ < 0.05; Gm9968: **$p$ < 0.0005, Student's t-test) (Supplementary Fig. 1K). These results confirm that D17Rik is enriched in the cytoplasm of pyramidal neurons. In addition, FISH analysis of D17Rik localization in cultured primary hippocampal neurons from mice showed punctate distribution in dendrites (Fig. 1J), indicating that D17Rik might be transported to dendrites.

## D17Rik is transported along dendrites and localized within dendritic spines

The dendritic localization of D17Rik suggests the possibility of a specific molecular motor-mediated transport process. We sought to directly test this by visualizing the dendritic transport of D17Rik. Based on its localization and modulation of its expression by experience, we named 2610035D17Rik lncRNA as SLAMR (*S*ynaptically *L*ocalized *A*ctivity *M*odulated lnc*R*NA).

Given that we are interested in visualizing the transport of a lncRNA, we could not rely on directly tagging the lncRNA with a fluorophore. Instead, we used the MS2-MCP system, which involves tagging the RNA of interest with multiple MS2 hairpin loops that will be recognized specifically by the MS2 Coat Protein (MCP) linked to a fluorophore[44–48]. We followed the protocols outlined by Bauer et al. 2017 and 2019 for optimal imaging of RNA transport in neurons. For these imaging experiments, we used rat primary hippocampal

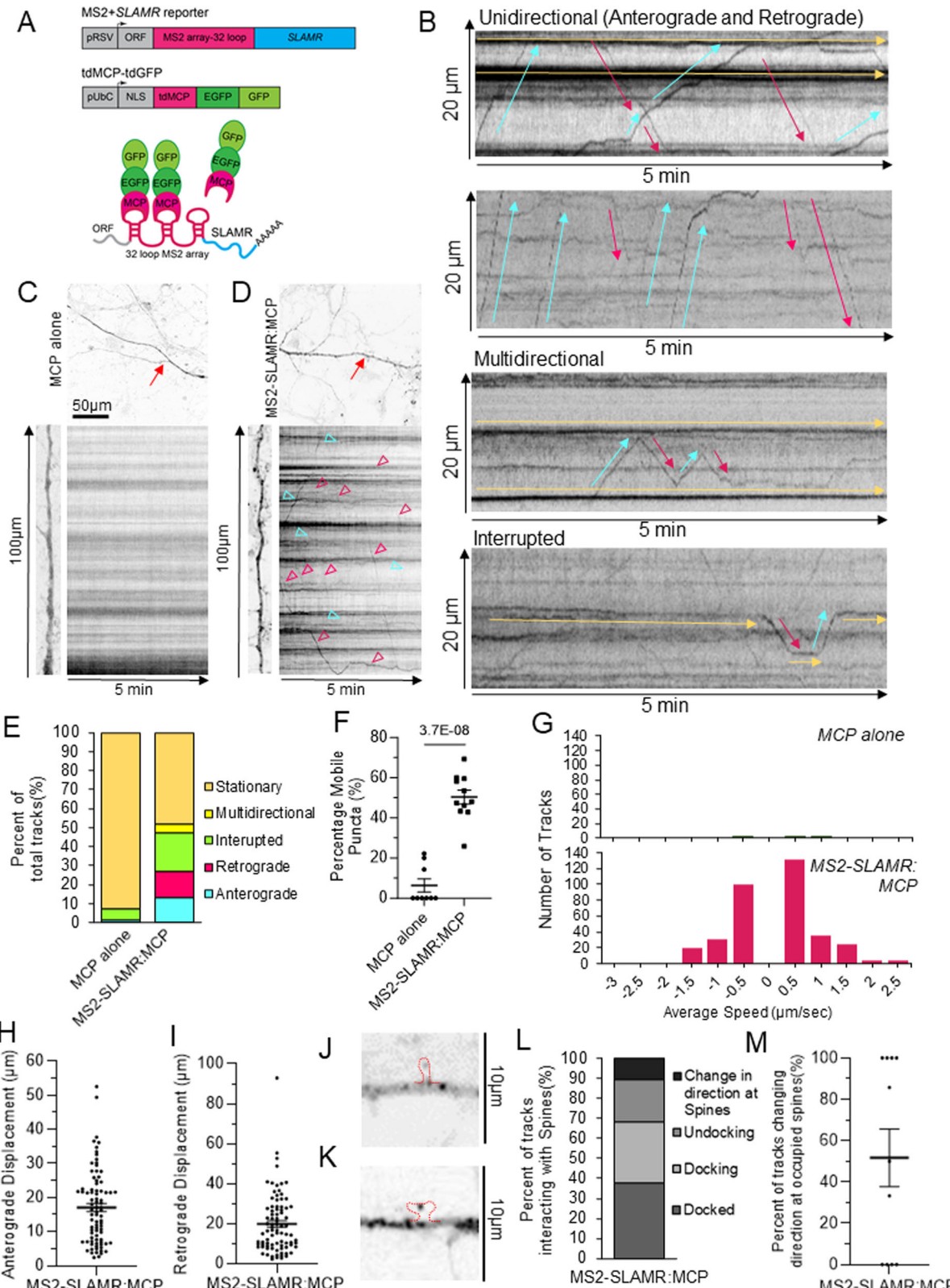

neuronal cultures, which are better suited to express MS2 with an elevated number of loops (32), for a greater signal-to-noise ratio, and tolerate double and triple transfections. GFP signal from MCP constructs alone is localized in the nucleus, but after MS2-SLAMR tandem synthesis, MCP binds MS2 loops in the cytoplasm, allowing the visualization of SLAMR transport via GFP positive particles traveling through the neuron (Fig. 2A). Despite the presence of some MCP

particles outside the nucleus, FISH labeling of SLAMR demonstrated the co-localization of MS2-MCP constructs in both cytoplasm and dendrites (Supplementary Fig. S2A). These results, confirm we are reliably detecting SLAMR lncRNA and allowing for the visualization of its transport in living neurons.

Next, we investigated the dynamics of SLAMR transport through spinning disc confocal microscopy and analysis of single neurons

**Fig. 2 | MS2-SLAMR displays directed dendritic transport in rat hippocampal neurons. A** Scheme of MS2-SLAMR reporter constructs and tdMCP-GFP expression cassettes and the MS2 system. pRSV Rous sarcoma virus promoter, pUBC Ubiquitin C promoter, ORF open reading frame, NLS nuclear localization signal, tdMCP tandem MS2 coat protein. **B** Representative kymographs illustrating differences in unidirectional MS2-SLAMR mRNA granule transport speed, displacement, directionality, multidirectional transport, and interrupted transport. Selected anterograde, retrograde, or stationary tracks are indicated in blue, pink, or yellow arrows, respectively. **C, D** Top and left: dendritic branch which the kymograph and movies came from. Red arrows=selected branch. Right: representative kymograph from neuron transfected MCP alone (**C**) or MS2-SLAMR (**D**) and MCP. Soma is orientated at the bottom; the dendrite end is towards the top. Distance progresses along the vertical axis, time progresses along the horizontal axis. Pink arrowheads=retrograde tracks. Blue arrows=anterograde tracks. **E** Quantification of transport dynamics of MS2 only and MS2-SLAMR in 5-min time-series acquisitions. **F.** Percent of mobile MCP alone or MS2-SLAMR granule. Individual values shown. (MCP alone $n$ = 9, MS2-SLAMR:MCP $n$ = 11 neurons; ****p-value < 0.0001, Two-tailed Student's t-test; mean ± SEM. **G** Distribution of speeds of MS2-SLAMR and MCP alone in primary hippocampal neurons. **H** Anterograde displacement of MS2-SLAMR. Individual tracks shown. ($n$ = 85 tracks from 11 neurons; mean ± SEM). **I** Retrograde displacement of MS2-SLAMR. Individual tracks shown. ($n$ = 85 tracks from 11 neurons; mean ± SEM). **J** MS2-SLAMR:MCP granule in a thin spine. Red outline of spine based on red channel RCaMP1.07. **K** MS2-SLAMR:MCP granule in a mushroom spine. Red outline of spine based on red channel RCaMP1.07. **L** Distribution of MS2-SLAMR dynamics in relation to dendritic spines (labeled with PSD95-mCherry). Docked: granule stays within 2.5 μm of PSD95 during the recording. Docking: granule moves to and then resides within 2.5 μm of PSD95 during the recording. Undocking: a formerly stationary granule that was within 2.5 μm of PSD95 travels away from PSD95. Change in direction at spines: a mobile granule that reverses the direction it was moving within 2.5 μm of PSD95. **M** Percent of MS2-SLAMR granules changing direction at a dendritic spine already occupied by MS2-SLAMR. Results from individual neurons shown. ($n$ = 11 neurons; mean ± SEM).

expressing the MS2 system. We recorded each neuron for 5 minutes at ~1 fps (frames per second) and analyzed the trafficking of single RNA granules. Kymographs of dendritic regions were acquired at a distance of more than 10 μm from the soma, and single trajectories were identified (Fig. 2B–D, Supplementary Fig. S2 B, C, Supplementary Movies S1–2). These experiments revealed different RNA transport patterns, even for the same granules, in a single recording as previously found for mRNAs following the sushi-belt model[47]. The analysis of the underlying frequencies found that less than 50% of the RNA particles remained stationary during the acquisition period (Fig. 2E). This fraction is significantly lower than granules in the control condition (MCP-alone) where more than 90% remain stationary (Fig. 2E). Furthermore, more than 50% of granules traversed the dendrites in a highly dynamic manner, displaying multidirectional, interrupted, retrograde, and anterograde movements, respectively (MCP-alone $n$ = 9 neurons, MS2-SLAMR:MCP $n$ = 11 neurons; Student's t-test; Fig. 2E, F) in a 5-min period of acquisition. While MS2-SLAMR and MCP-alone granules on average displayed similar anterograde and retrograde velocities (Anterograde: MCP-alone = $0.56 \pm 0.17$ μm/s, MS2-SLAMR:MCP = $0.56 \pm 0.04$ μm/s) and (Retrograde: MCP-alone = $0.37 \pm 0.12$ μm/s, MS2-SLAMR:MCP = $0.49 \pm 0.03$ μm/s) there were only 4 total tracks to measure in the MCP-alone condition compared to 352 tracks of MS2-SLAMR:MCP ($n$ = 9 dendrites per condition, Student's t-test, Fig. 2G). Due to the very low number of tracks in the MCP-alone condition preventing us from making meaningful statistical comparisons with MS2-SLAMR: MCP, we focused on analyzing MS2-SLAMR dynamics. MS2-SLAMR showed varied anterograde and retrograde displacement ($n$ = 9 neurons, Fig. 2H, I). These results demonstrate that SLAMR is actively similarly transported along the dendrites to previously reported mRNAs required for local protein synthesis involved in plasticity processes[47,48].

Interestingly, we also observed that MS2-SLAMR could enter a variety of dendritic spine compartments. Specifically, we noticed that MS2-SLAMR:MCP granules are indeed also found within thin and mushroom spines (Fig. 2J, K). To further characterize MS2-SLAMR's behavior in relation to dendritic spines, we imaged MS2-SLAMR concurrently with PSD95-mCherry to ensure we were observing its transport specifically in dendrites using rat primary neuronal cultures. We found that MS2-SLAMR RNA granules are frequently docked at, undergo docking or undocking, or change direction at dendritic spines labeled with PSD95 (Fig. 2L, Supplementary Fig. S2D, E, Supplementary Movie S3). Interestingly, slightly over 50% of the time, MS2-SLAMR changed direction at a spine that was already occupied by an MS2-SLAMR granule ($n$ = 11 dendrites, Fig. 2M). This observation suggests that there may be a mechanism determining whether a given neuron already contains 'enough' SLAMR within a spine and thus needs to direct SLAMR to an unoccupied spine.

## SLAMR is actively transported along dendrites via KIF5C
As the dynamics of MS2-SLAMR suggest directed motor-dependent transport, our next objective was to investigate the kinesins involved in SLAMR transport. We first tested three kinesins that participate in dendritic transport: KIF2A, KIF5C, and KIF11[8,13,49,50]. We used siRNAs to silence these kinesins in primary mouse hippocampal neuronal cultures under basal conditions. RT-qPCR results from total homogenates and synaptoneurosomes revealed SLAMR expression significantly changes when transport by KIF2A, KIF5C, and KIF11 is individually disrupted ($n$ = 3 per group, *$p$ < 0.05, **$p$ < 0.005. One-way ANOVA followed by Dunnett's test, Fig. 3A). Specifically, the silencing of *Kif2a* significantly increased SLAMR in the synaptic fractions, indicating a possible role of this kinesin as a negative regulator of SLAMR transport to the synapse. Silencing *Kif5c* decreased SLAMR in the homogenate and the synaptic fraction, suggesting that it positively regulates the global abundance of SLAMR including the synapses. Silencing *Kif11* only decreased SLAMR in the total homogenate, indicating that it is correlated with SLAMR expression but likely does not participate in its localization to synapses. Furthermore, overexpression of KIF5C resulted in an increase in the expression of SLAMR in homogenates as well as in synaptoneurosome fractions ($n$ = 3, $p$ < 0.05, student's t-test; Supplementary Data S3A). Taken together, these results identify KIF5C as a potential motor for the dendritic transport of SLAMR. To further investigate this possibility, we examined the dynamics of MS2-SLAMR basal transport in KIF5C silenced neurons.

Briefly, primary rat hippocampal neurons were transfected with MS2-SLAMR:MCP-RFP and either a scrambled (Scr-shRNA) control or the *Kif5c*-shRNA construct that leads to a significant decrease in KIF5C abundance in hippocampal neurons[8,50] (Fig. 3B). We then proceeded with spinning disc confocal microscopy of neurons co-expressing the MS2-SLAMR:MCP-RFP RNA imaging reporter and the indicated shRNA (resulting in eGFP expression). We recorded each neuron for 5 minutes at ~1 fps. For this analysis, we focused on 100 μm dendritic regions where we observed robust trafficking of single RNA granules >10 μm away from the soma and generated kymographs (Supplementary Fig. S3A–D, Supplementary Movies S4). Loss-of-function of KIF5C significantly reduced the anterograde and retrograde velocities of MS2-SLAMR RNA granules compared to the Scr-shRNA control ($n$ = 10 per condition, Student's t-test, exact $p$-values shown, Fig. 3C, D, Supplementary Data S3C, D). Additionally, KIF5C loss-of-function significantly reduced the distance traveled by MS2-SLAMR granules from the soma when compared to the Scr-shRNA control, in both anterograde and retrograde directions ($n$ = 10 per condition, exact $p$-values shown, Student's t-test; Fig. 3E, F). Lastly, *Kif5c* knockdown significantly decreased the percentage of mobile MS2-SLAMR granules to $11.9 \pm 2.3\%$ compared to $35.7 \pm 3.8\%$ mobile particles in the Scr-shRNA control ($n$ = 10 per condition, exact $p$-values shown,

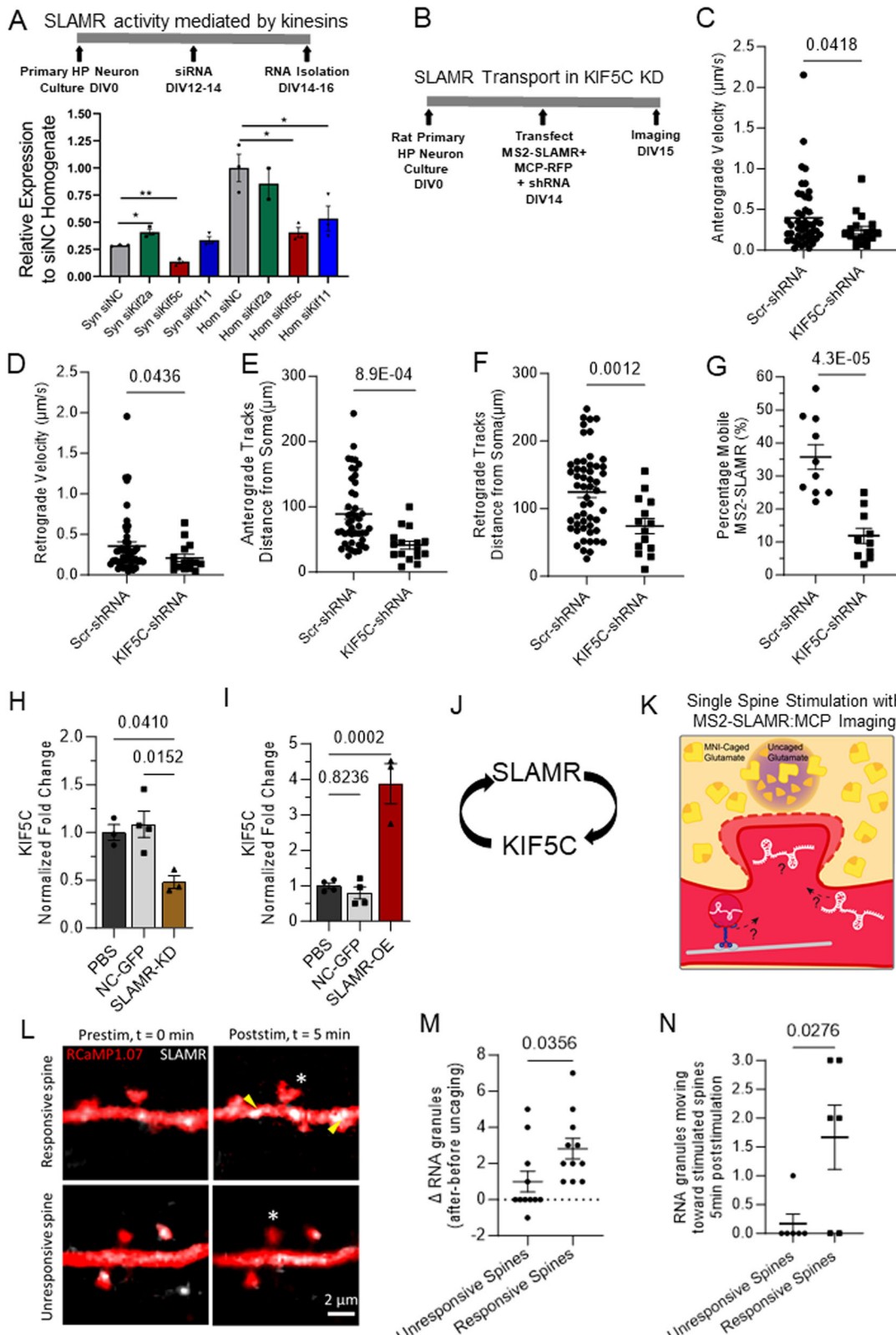

**Nature Communications** | (2024)15:2694

Student's t-test, Fig. 3G). Taken together, these results demonstrate that SLAMR depends on KIF5C for transport along dendrites.

As KIF5C is crucial for SLAMR transport and localization, we considered the possibility of a reciprocal regulation between these two molecular factors. To elucidate this, we assessed whether the expression of KIF5C is modulated by SLAMR. Therefore, we studied the effect of both the loss- and gain-of-function of SLAMR on KIF5C abundance in

hippocampal neurons. Specifically, we treated primary mouse hippocampal neurons with PBS or transduced them with lentiviral particles of negative control (NC-GFP), knockdown (shSLAMR), or overexpression (OE-SLAMR) constructs of SLAMR. After extracting RNA from these cultures, and evaluating SLAMR expression to confirm knockdown and overexpression (Supplementary Fig. S3E, F), we then examined the abundance of KIF5C. Interestingly, loss-of-function of

**Fig. 3 | SLAMR and kinesin KIF5C reciprocally regulate each other. A** Top: Experimental timeline. Bottom: qRT-PCR results show changes in SLAMR RNA levels in total homogenates and synaptic fractions of primary mouse neurons after kinesin silencing ($n = 3$ biological replicates for each except Hom siKif2a $n = 2$; *$p = 0.0184$, **$p = 0.0002$; One-way ANOVA + Dunnett's test; mean $\pm$ SEM). **B** Experimental timeline for KIF5C knockdown and MS2-SLAMR:MCP-RFP transport in rat hippocampal neurons. **C, D** Anterograde and retrograde velocity of MS2-SLAMR in KIF5C-KD or scrambled control neurons (N = 10 neurons each; Two-tailed Student's t-test; mean $\pm$ SEM). **E, F** The distance from the soma that anterograde (**E**) and retrograde tracks (**F**) of MS2-SLAMR begin transport in KIF5C-KD and scrambled control neurons. (N = 10 neurons each; Two-tailed Student's t-test; mean $\pm$ SEM). **G** The percentage of mobile MS2-SLAMR:MCP-RFP in KIF5C-KD and scrambled control neurons (N = 10 neurons each; Two-tailed Student's t-test; mean $\pm$ SEM). **H** Normalized fold change of Kif5c mRNA abundance in mouse neuronal cultures transduced by shSLAMR, NC-GFP lentivirus, or treatment with PBS. (PBS and SLAMR-KD $n = 3$, NC-GFP $n = 4$ biological replicates; One-way ANOVA + Tukey's test; mean $\pm$ SEM). **I** Normalized fold change of Kif5c mRNA abundance in mouse

neuronal cultures transduced by lentivirus containing SLAMR-OE compared to NC-GFP or treatment with PBS. (PBS and NC-GFP $n = 4$, SLAMR-OE $n = 3$; One-way ANOVA + Tukey's test; mean $\pm$ SEM). **J** Model of the reciprocal regulation of SLAMR and KIF5C. **K** Diagram illustrating single spine stimulation DIV18-21 rat neurons transfected with RCaMP1.07 (red) and MS2-SLAMR:MCP (white). Individual spines were stimulated with 30 pulses of 2-photon excitation to uncage MNI-Caged glutamate and evaluate transport dynamics of SLAMR to dendritic spines. **L** Single frames of an unresponsive spine with no increase in volume and a responsive spine that did grow following local glutamate uncaging, prestimulation, and 5 min poststimulation. Red=RCaMP1.07, White=MS2-SLAMR:MCP granules. White asterisk= stimulated spine. **M** The change in number of MS2-SLAMR:MCP granules prestimulation to 5 min poststimulation in a 5 μm dendritic region of the stimulated spine. ($n = 11$ neurons each; Two-tailed Student's t-test; mean $\pm$ SEM). **N** The change in number of MS2-SLAMR:MCP granules moving toward the stimulated spine within 5 min after stimulation in a 25 μm region of the stimulated spine ($n = 6$ neurons each; Two-tailed Student's t-test; mean $\pm$ SEM).

SLAMR diminished the level of KIF5C to $0.48 \pm 0.07$ fold compared to the NC-GFP control ($n = 4$ per condition; one-way ANOVA followed by Tukey's test; Fig. 3H) while gain-of-function of SLAMR increased KIF5C by $3.877 \pm 0.56$ fold compared to NC-GFP control ($n = 4$ for PBS, and NC-GFP, $n = 3$ for OE-SLAMR; one-way ANOVA followed by Tukey's test; Fig. 3I). Together, these results suggest that SLAMR and KIF5C reciprocally regulate each other's expression in primary mouse hippocampal neurons (Fig. 3J).

## SLAMR is recruited to stimulated spines

As SLAMR expression increases in the hippocampus following CFC, is actively transported along dendrites, and is localized within spine compartments, it led us to consider the possibility that SLAMR might be recruited to spines in response to stimulation for activity-dependent structural changes. Supporting this idea, a few other RNAs have been demonstrated to increase in dendrites and dendritic spines following stimulation[13,47]. Thus, we next examined whether local spine stimulation impacts SLAMR localization within spines in hippocampal neurons. To test the recruitment of SLAMR to activated spines, we used two-photon (2p) glutamate uncaging to stimulate individual dendritic spines in primary rat hippocampal neurons expressing MS2-SLAMR:MCP and RCaMP1.07 (volume marker)[51] (Fig. 3K). We established a criterion for 'responsive' spines, based on previous studies, where they must increase in size by at least 10% of their volume upon stimulation[52,53]. We then compared the recruitment of SLAMR to spines that were responsive to those that were unresponsive ($n = 11$ unresponsive spines, $n = 11$ responsive spines exact $p$-values shown, two-way ANOVA + Sidak's multiple comparison test; Fig. 3L, Supplementary Fig. S3G). We found that the responsive spines had an increase of ~3 RNA granules within a 5 μm dendritic region of the stimulated spine compared to a modest recruitment of ~1 RNA granule near nonresponsive spines, 5 minutes after stimulation (based on previous studies)[13,52] (Nonresponsive $n = 11$ spines, Responsive $n = 11$ spines, Student's t-test; Fig. 3L, M). We next examined the direction in which SLAMR was moving in a 25 μm dendritic region of the stimulated spines following local glutamate uncaging. Here, we found that in the 5 min following stimulation, the responsive spines showed ~2 RNA granules moving toward the stimulated spine, while the non-responsive spines showed ~0.2 RNA granules moving toward the stimulated spine (Nonresponsive $n = 6$ spines, Responsive $n = 6$ spines; Student's t-test; Fig. 3N, Supplementary Fig. S3H, I, Supplementary Movie S5). Together, these results demonstrate the selective recruitment of SLAMR to spines exhibiting structural plasticity.

## SLAMR is required for arborization and activity-dependent structural plasticity

After we established that SLAMR is transported along dendrites and is recruited to spines in an activity-dependent manner, we next sought to

examine if SLAMR functions broadly in regulating the morphology of the dendritic arbor and spines. To do so, we carried out loss-of-function experiments by shRNA KD of SLAMR to determine the necessity of its expression for dendritic arborization and synapse function in primary mouse hippocampal neuronal cultures. We used a shRNA plasmid to constitutively silence the expression of SLAMR and, alternatively, one under a doxycycline-inducible promoter (TET) for temporal control of SLAMR KD. We used a plasmid expressing a scrambled sequence as a negative control (NC). All plasmids also expressed eGFP to visualize transfected neurons and assess morphological changes (Fig. 4A, B). Sholl analysis of the dendritic arbor revealed that both plasmids for SLAMR silencing (shRNA and TET-ON condition) induced a significant decrease in dendritic arborization of the transfected neurons compared to the NC (NC-GFP $n = 9$, SLAMR-KD $n = 10$, TET-SLAMR $n = 11$; Two-way ANOVA + Tukey's test; Fig. 4B, C). However, we did not find any significant changes in spine density and morphology with SLAMR knockdown compared to the control (Fig. 4D–F).

Though SLAMR loss-of-function did not affect spine number per 100 μm dendritic length, we were curious about spine function as SLAMR is recruited to stimulated spines exhibiting structural plasticity. Therefore, we investigated the effect of SLAMR KD upon induction of structural long-term potentiation (sLTP) in single dendritic spines by local glutamate uncaging, a model for functional synaptic plasticity and learning[52]. We employed shRNA plasmids targeting SLAMR (TET-SLAMR) or NC-GFP (Fig. 4G) and assessed activity-dependent changes in spine morphology using primary mouse hippocampal neurons. sLTP was induced in single dendritic spines by local glutamate uncaging with 2p excitation. This uncaging stimulus induced a large transient volume increase ($49.9 \pm 11.8\%$ at +1.5 min after the first uncaging) at the stimulated spine expressing the control shRNA ($n = 12$ spines from 4 cells) (Fig. 4H–J). However, the spine enlargement was significantly suppressed ($15.6 \pm 9.3\%$ at +1.5 min after the first uncaging) at a transient phase in neurons that inhibited endogenous SLAMR ($n = 13$ spines from 5 cells; $p = 0.03$, Two-way ANOVA + Tukey's test; Fig. 4 I–J). Interestingly, a sustained sLTP (+28.5-32.5 min after the first uncaging) was induced and maintained not only in the control group ($10.3 \pm 0.4\%$) ($p < 0.001$; One-way ANOVA + Tukey's test) but also in SLAMR KD group ($14.5 \pm 0.8\%$) ($p < 0.001$, Two-way ANOVA + Tukey's test; Fig. 4I-J). The degree of sustained spine growth was comparable between each group (NC $9.8 \pm 6.9\%$ and SLAMR KD $14.0 \pm 12.6\%$ at +30.5 min) ($p = 0.97$, Two-way ANOVA + Tukey's test; Fig. 4K). To examine whether endogenous SLAMR is responsible for the sensitivity of postsynaptic structural remodeling, fast rate imaging (3.91 Hz compared to 0.65 Hz for slow rate) was performed during glutamate uncaging. Spine growth was induced during a train of glutamate uncaging and kept increasing immediately

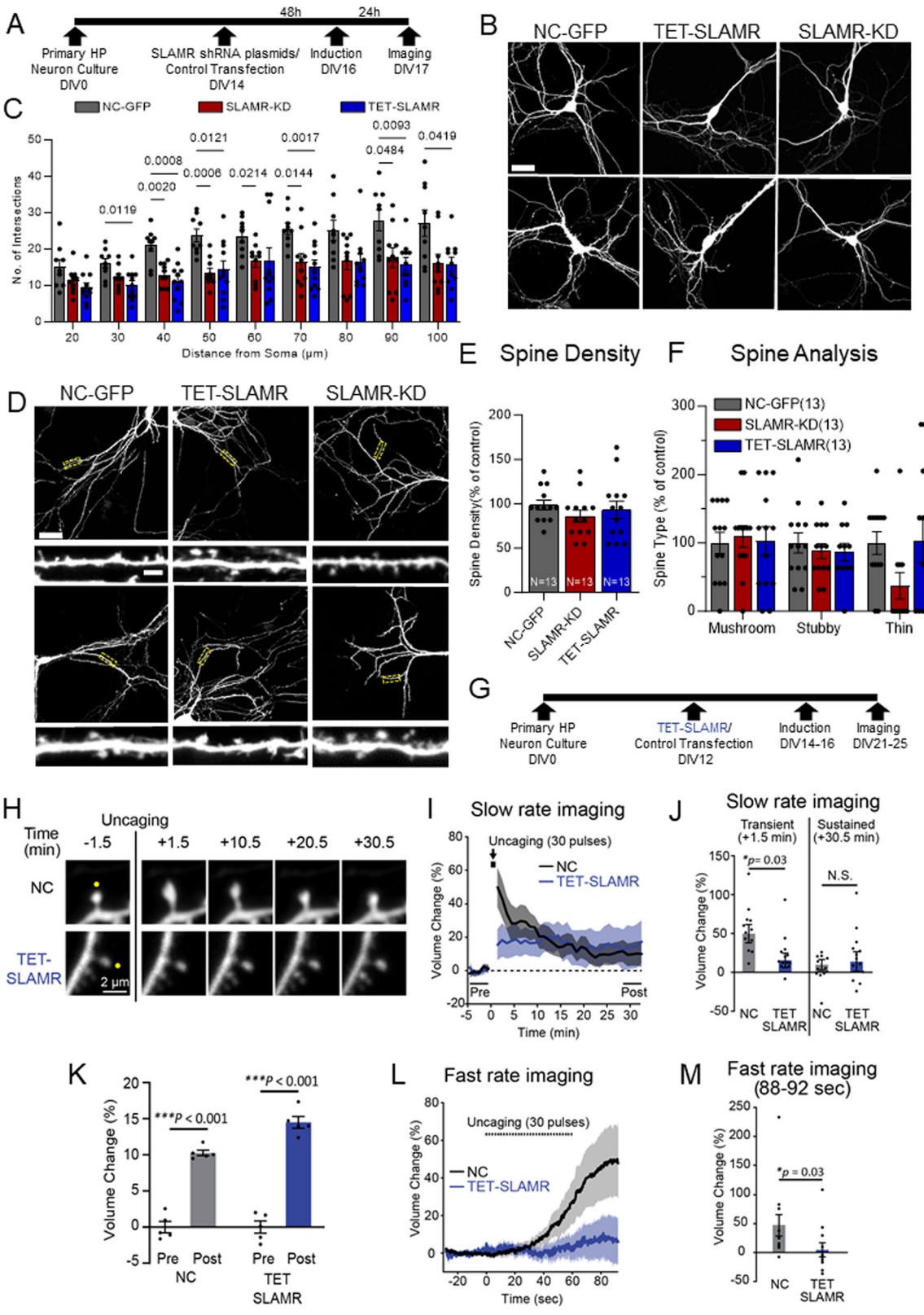

after the stimulation in the control group (48.8 ± 18.8% between 88-92 sec after the 1st uncaging) ($n = 12$ spines from 4 cells; Fig. 4L and M). However, the initiation of spine growth was not observed during uncaging, and spine enlargement was significantly attenuated in the SLAMR KD group (6.7 ± 12.3% between 88-92 sec after the first uncaging) ($n = 11$ spines from 4 cells; $p = 0.03$, Mann-Whitney's U test; Fig. 4L–M). Taken together, we conclude from these results that

SLAMR is involved in the transient structural plasticity of spines but is not critical for sustained sLTP.

## SLAMR gain-of-function enhances arborization, spine density, and translation

We next asked whether enhancing the expression of SLAMR would produce an enhancement in dendritic arborization and spine density.

**Fig. 4 | SLAMR facilitates dendritic arborization and transient structural plasticity in mouse hippocampal neurons. A** Experimental timeline for morphology studies. **B** Confocal projection images of hippocampal neurons. Scale Bar = 40 μm. **C** Quantification of dendritic morphology using Sholl analysis of intersections per 10-μm step size. Changes compared between SLAMR constitutive and inducible (TET) KD and NC-GFP(negative control) (NC-GFP $n = 9$, SLAMR-KD $n = 10$, TET-SLAMR $n = 11$ neurons; Two-way ANOVA + Tukey's test; mean ± SEM). **D** Confocal projection images showing region analyzed for spine morphology and enlarged image in the inset for spine details. Cell Body Scale Bar=40 μm. Dendrite inset Scale bar=2 μm. **E, F** SLAMR KD did not induce significant changes in the total number of spines or their different subtypes. ($n = 13$ dendrites each; **E** One-way ANOVA. **F** Two-way ANOVA + Tukey's Test; mean ± SEM). **G** Experimental timeline

of hippocampal primary culture, inducible shRNA plasmid transfection, and doxycycline treatment for two-photon imaging with glutamate uncaging. **H** Representative fluorescence images of SLAMR shRNA-EGFP (TET-SLAMR) and scrambled shRNA-EGFP (NC-GFP) before and after glutamate uncaging pulse (30 pulses, 0.5 Hz). Scale bar, 2 μm. **H–K** Glutamate uncaging analysis of spine changes in volume with slow rate imaging (**I**) shows a significant decrease in TET-SLAMR condition compared with NC-GFP control at the transient phase (**J**) (NC $n = 12$ neurons, TET-SLAMR $n = 13$ neurons; Two-way or One-way ANOVA +Tukey's test; (**K**) NC pre v. post ***$p = 1.65E-07$, TET-SLAMR pre v. post ***$p = 1.18E-09$; mean ± SEM). **L, M** Fast rate analysis of the spine volume during glutamate uncaging (NC $n = 12$ neurons, TET-SLAMR $n = 11$ neurons; Two-tailed Mann-Whitney U test, mean ± SEM).

Therefore, we carried out gain-of-function experiments by transfecting primary mouse hippocampal neurons with a plasmid expressing full-length SLAMR (SLAMR-OE) under the control of a CMV promoter. We also included an empty backbone plasmid as negative control (NC-GFP) (Fig. 5A). Sholl analysis of intersections shows that compared to NC-GFP, SLAMR-OE induces a significant increase in the number of branches (NC-GFP, $n = 15$, SLAMR-OE, $n = 14$; Two-way ANOVA + Tukey's test; Fig. 5B, C). Additionally, we found an increase in spine density and the percentage of thin and mushroom spines in neurons transfected with SLAMR-OE compared to the NC-GFP condition (NC-GFP, $n = 15$, SLAMR OE, $n = 17$; Student's t-test (E) and Two-way ANOVA + Šídák's multiple comparisons test (F); Fig. 5D–F). These results, together with the loss-of-function experiments, indicate that SLAMR is a key mediator of dendritic arborization and spine morphology.

The observation that SLAMR-OE was sufficient to induce an increase in arborization as well as the reciprocal regulation of SLAMR and KIF5C, suggested that SLAMR may influence translational changes critical for neuronal outgrowth in hippocampal neurons Previously, we reported that KIF5C-OE produced an increase in translation in primary mouse hippocampal neurons[8]. Thus, we investigated whether changes in SLAMR levels similarly affected translation. First, using mouse primary hippocampal neuronal culture, we carried out puromycin labeling of newly synthesized proteins in NC-GFP and SLAMR-OE neurons, followed by immunostaining. SLAMR-OE resulted in increased staining of puromycin-labeled proteins in the cell body (SLAMR-OE = 1643875 ± 257055 CTCF, $n = 6$), compared to the NC-GFP control (NC-GFP = 1055403.278 ± 57712.63243 CTCF, $n = 4$) (unpaired Student's t-test; Fig. 5G, H, Supplementary Fig. S4A). Such changes were also observed in dendrites (SLAMR-OE = 1339079 ± 226821. CTCF, $n = 8$) when compared to NC-GFP control (NC-GFP = 581139. ± 45647 CTCF, $n = 6$) (unpaired two-tailed Student's t-test; Fig. 5I). These results indicate that SLAMR likely participates in the global regulation of translation in neurons.

To better examine the role of SLAMR in global translation, we decided to perform a complementary experiment by asking whether a reduction in SLAMR levels conversely reduces translation. Likewise, we carried out puromycin labeling of newly synthesized proteins in NC-GFP and SLAMR-KD primary mouse hippocampal neurons, followed by immunostaining. SLAMR-KD resulted in no statistically significant change in the staining of puromycin-labeled proteins in the cell body (SLAMR-KD = 15983 ± 1981 CTCF, $n = 13$), compared to the NC-GFP control (NC-GFP = 20705 ± 2164 CTCF, $n = 11$) ($p = 0.1206$, unpaired Student's t-test; Fig. 5J, Supplementary Fig. S4B). However, SLAMR-KD did cause a strong reduction in staining of puromycin in dendrites (SLAMR -KD = 1711 ± 304 CTCF, $n = 18$) when compared to NC-GFP control (NC-GFP = 3492 ± 541 CTCF, $n = 6$) (**$p < 0.01$, unpaired two-tailed Student's t-test; Fig. 5K, Supplementary Fig. S4B). These results support that SLAMR is implicated in the global regulation of translation in neurons, particularly in dendrites.

To further characterize the role of SLAMR in modulating neuronal translation, and considering its synaptic localization, we next evaluated whether the loss-of-function of SLAMR diminishes local

translation at the synapse. Thus, we isolated synaptoneurosomes from mouse primary hippocampal neurons transduced with lentivirus containing scrambled shRNA (shScr) or SLAMR shRNA (shSLAMR), respectively, and probed for the presence of the translation machinery components (Fig. 5L). We first validated the isolation of synaptoneurosomes by probing for synaptic marker glutamate receptor (GluR2) and Synaptophysin in total homogenate, cytosolic, and synaptoneurosome fractions. We found GluR2 and Synaptophysin to be highly enriched in the synaptoneurosome fraction compared to the total homogenate and cytosolic fraction ($n = 3$ per condition; One-way ANOVA + Tukey's test; Supplementary Fig. S4C–E). Western blot analysis of synaptoneurosomes relative to shScr, showed shSLAMR significantly decreased markers for protein synthesis such as eIF3g (shSLAMR =0.21 ± 0.03, shScr=0.36 ± 0.05, $n = 3$), eIF2a (shSLAMR =0.07 ± 0.02, shScr=0.14 ± 0.03, $n = 3$), and P70S6K (shSLAMR =0.21 ± 0.06, shScr= 0.28 ± 0.07, $n = 3$) (unpaired two-tailed Student's t-test; Fig. 5L–O). These results reveal a significant correlation between key markers of protein synthesis and SLAMR function, suggesting that SLAMR mediates translation in hippocampal neurons globally and within the synapse.

## SLAMR interacts with specific lncRNAs, miRNAs, mRNAs, and proteins

LncRNAs typically participate in cellular processes by acting as part of a complex with other RNAs and proteins. Therefore, to gain deeper insight into the mechanism of SLAMR function, we isolated SLAMR-containing complexes from the intact hippocampus of naive mice and analyzed the associated RNA and protein components by total RNA-seq, small RNAseq, and proteomics. Briefly, we used a biotinylated probe containing the full-length sense sequence of SLAMR (Fig. 6A), or a biotinylated antisense strand of SLAMR as a negative control (Supplementary Fig. S5A) to pulldown SLAMR-associated complexes. qRT-PCR analysis of isolated complexes indicated significant enrichment of SLAMR, supporting the efficiency of SLAMR pulldown (*$p$-value < 0.05, Student's t-test, Supplementary Fig. S5B). Silver staining following SDS-PAGE analysis of eluted complexes revealed distinct bands for proteins between 35 to 64 KDa (Supplementary Fig. S5C). These purified complexes were processed for both RNA and protein isolation. The RNA samples were submitted for total RNAseq and small RNAseq, while the proteins were analyzed using liquid chromatography-mass spectrometry (LC-MS/MS). This analysis resulted in the identification of 271 coding and non-coding transcripts enriched in the sense condition by DEseq analysis compared to the antisense condition ($p$-value < 0.05) (Fig. 6B). From these transcripts, 10 miRNAs were identified (Fig. 6C and Supplementary Data S6C), most of them with unknown functions (eg. Gm44355, Gm24049, Gm22234), except for mir3064 and mir1839 that are involved in post-transcriptional and translational regulation[54]. We next used Metascape to analyze the list of mRNAs significantly enriched in the sense condition (Fig. 6D, Supplementary Data S6D). Results from this analysis showed that most of the mRNAs that interact with SLAMR are involved in functions like the electron transport chain, regulation of ion transmembrane transport, transsynaptic signaling, or

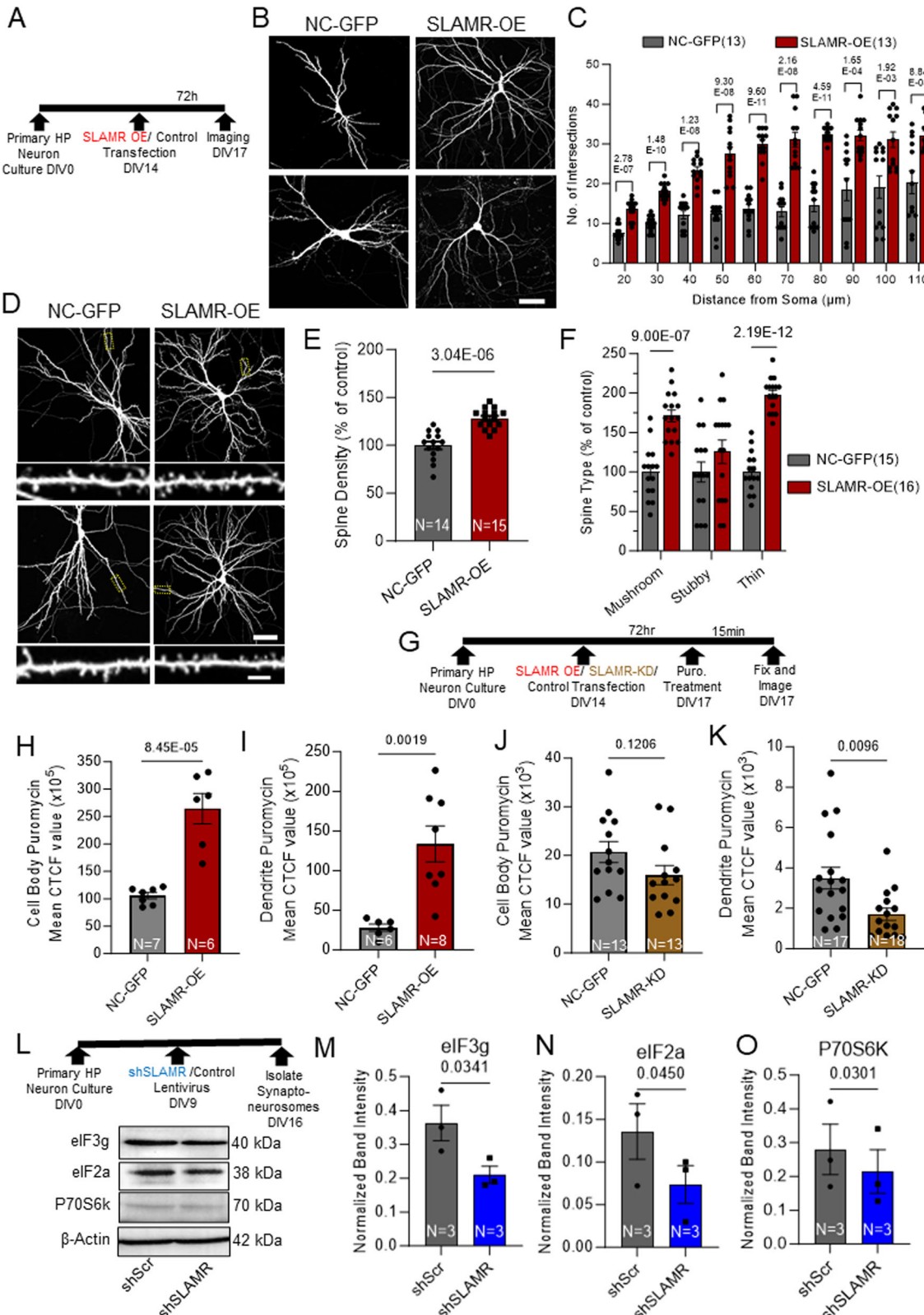

positive regulation of excitatory postsynaptic potential, indicating a possible relation of this lncRNA in mitochondrial function and synaptic transmission. However, this possibility must be explored in future studies to draw stronger conclusions.

Despite the presence of some common proteins detected by LC-MS/MS in both conditions (sense and antisense), the proteomic analysis identified 38 unique proteins in the sense condition (Fig. 6E). They were found to be involved in diverse functions, including mitochondrial function (ex. ATP synthase subunit β), transcription regulation (ex. Histone H2A), or cytoskeleton neurofilaments (Supplementary Data S6E). Considering the possible limitations of the LC-MS/MS results, we decided to focus on two specific proteins that were unique in the sense condition as well as validated with consistency by western blot (WB): the Calcium-calmodulin protein kinase alpha (CaMKIIα) and

**Fig. 5 | Gain-of-function of SLAMR enhances dendritic arborization, spine density, and local protein synthesis in mouse hippocampal cultures.**
**A** Experimental timeline for morphology studies. **B** Confocal projection images of hippocampal neurons. Scale Bar=40 μm. **C** Quantification of dendritic morphology changes using Sholl analysis of intersections per 10-μm step size. Changes compared between NC-GFP and SLAMR-OE (*n* = 13 neurons each; Two-way ANOVA + Tukey's test; mean ± SEM). **D** Confocal projection images showing region analyzed for spine morphology and digitally enlarged images in the inset for spine details. Cell Body Scale Bar=40 μm. Dendrite inset Scale bar=2 μm. **E, F** Spine density (two-tailed student's t-test; mean ± SEM) and type (Two-way ANOVA + Šídák's multiple comparisons test; mean ± SEM) in NC-GFP and SLAMR-OE neurons. **G** Experimental timeline for puromycin experiment. **H, I** Quantification of puromycin staining in

Supplementary Fig. S4A based on Corrected Total Cell Fluorescence (CTCF) in the cell body (**H**) and dendrites (**I**) of NC-GFP and SLAMR-OE neurons (two-tailed student's t-test; mean ± SEM). **J, K** Quantification of puromycin staining in Supplementary Fig. S4B based on CTCF in the cell body (**J**) and dendrites (**K**) of NC-GFP and SLAMR-KD neurons decrease (two-tailed student's t-test; mean ± SEM). **L** Top: Experimental timeline for Synaptoneurosome isolation. bottom: Loss-of-function of SLAMR results in diminished components of local translation machinery. Western blot analysis of β-actin, P70S6K, eIF2a, eIF3g, isolated from primary hippocampal neurons transduced by either shScr or SLAMR lentivirus synaptoneurosomes. **M–O** Quantification of (**L**) normalized to β-actin (two-tailed student's t-test). Data are presented as mean ± SEM.

the intermediate filament Vimentin. LC-MS/MS experiment identified 5 unique peptides for CaMKIIα (Supplementary Fig. S5D) that were also validated by WB through 4 independent pull-down experiments with 3 replicates each (*n* = 12 total; ***$p < 0.0005$, unpaired Student's t-test; Fig. 6F, G). It is well known that CaMKIIα plays an essential role in excitatory synaptic transmission and plasticity and that phosphorylation of CaMKII T282 is a prerequisite for this function[33,55]. So, we were curious if SLAMR had a role in the phosphorylation of CaMKII within the synapse. Therefore, we isolated synaptoneurosomes from primary mouse hippocampal neuronal cultures transfected with control shRNA (shScr) or SLAMR shRNA (shSLAMR) and probed for CaMKII and phosphorylated (T282) CaMKII (Fig. 6H). Reducing SLAMR in neuronal cultures led to a significant reduction in the ratio of phosphorylated to total CaMKII in synaptoneurosomes (*n* = 3 per condition; *$p < 0.05$, Student's paired t-test; Fig. 6I, Supplementary Data S5I). These results support a role for SLAMR in modulating CaMKIIα phosphorylation.

During the SLAMR-Biotin pulldown, we were also able to identify 20 unique peptides for Vimentin by LC-MS/MS (Supplementary Fig. S5E). The immunoblot analysis for Vimentin validated its enrichment in the sense condition in several different independent experiments (4×3 replicates, *n* = 11-12; ***$p < 0.0005$, unpaired Student's t-test; Fig. 6J, K). As vimentin is a key factor in CNS cell differentiation and neurite development[56,57], these findings suggest a possible role of SLAMR in plasticity processes and hippocampal-dependent memory. CaMKII and Vimentin are involved in different processes directly and indirectly related to learning and memory. The study of their interactions with SLAMR is crucial to understanding how this lncRNA is involved in hippocampal-mediated memory.

### SLAMR regulates spine plasticity and basal CaMKIIα activity in CA1 neurons

We next sought to investigate if SLAMR indeed has a role in modulating CaMKIIα activity in CA1 in basal and stimulated conditions. We utilized mouse organotypic hippocampal slice cultures co-expressing either a conditional SLAMR-KD construct (TET-SLAMR) or scrambled control (NC) and a CaMKIIα FLIM-FRET sensor of CaMKII activity[58] and measured both structural plasticity and CaMKII activity before and after spine stimulation by 2-photon glutamate uncaging (Fig. 7A). Similar to the structural plasticity analysis done in primary hippocampal cultures (Fig. 4H–M), we see a strong reduction in transient structural plasticity in TET-SLAMR CA1 pyramidal neurons compared to the NC control (NC-mCherry: N = 15 spines/5 neurons, TET-SLAMR: N = 17 spines/5 neurons; ****$p < 0.0001$, The Mann-Whitney U-test, Fig. 7B, D). Fluorescence lifetime imaging revealed CaMKIIα is locally activated by single spine glutamate uncaging (30 pulses, 0.5 Hz) (Fig. 7A), indicative of a reliable sensor and uncaging protocol. TET-SLAMR neurons displayed a significant reduction in CaMKII activity at basal levels (NC-mCherry: N = 15 spines/5 neurons, TET-SLAMR: N = 17 spines/5 neurons; ****$p < 0.0001$, The Mann-Whitney U-test, Fig. 7C-E). Due to this reduction in baseline CaMKII activity in SLAMR loss-of-function neurons, there was a significantly greater increase in the CaMKII activity (lifetime change) upon spine stimulation in TET-

SLAMR neurons compared to NC neurons (NC-mCherry: N = 15 spines/5 neurons, TET-SLAMR: N = 17 spines/5 neurons; Two-tailed unpaired Student's t-test (F). *$p < 0.05$, Fig. 7F). Together these data reveal that SLAMR regulates transient structural spine plasticity and basal CaMKIIα activity in CA1 pyramidal neurons.

To confirm if SLAMR can enhance CaMKII phosphorylation directly, we conducted an in vitro assay with isolated SLAMR, CaMKII, Calmodulin, and Ca$^{+2}$. For the SLAMR-mediated CaMKII activation assay, freshly prepared sense in vitro transcribed SLAMR RNA (slowly cooled, to allow for secondary structure folding) was diluted in kinase buffer (1:10) and added to CaMKII purified enzyme with varying concentrations of Calmodulin (0-25 ng/mL) (Fig. 7G). As shown in Fig. 7H, SLAMR sense RNA showed activation of CaMKII even in the absence of Calmodulin (0 ng/mL; No Template Control (NTC) = 0.79 ± 0.22, Sense=0.79 ± 0.22, Antisense=0.19 ± 0.08 Normalized 450 nm absorbance). Additionally, SLAMR showed greater CaMKII activation than both the NTC and antisense controls at 5 and 10 ng/mL Calmodulin (N = 4 biological replicates per condition; Two-way ANOVA + Tukey's test). There were no significant differences between sense SLAMR and NTC nor antisense SLAMR at 25 ng/mL, suggesting that at a certain concentration, Calmodulin can outcompete SLAMR binding to CaMKII. Interestingly, there were no statistically significant differences in CaMKII activation between NTC and antisense SLAMR at any calmodulin concentration, supporting our previous finding that antisense SLAMR does not bind CaMKII in hippocampal lysate (Fig. 6F,G). In summary, these results demonstrate that sense SLAMR can directly interact and activate CaMKII independently of Calmodulin.

### A 220-nucleotide fragment of SLAMR interacts with CaMKIIα and Vimentin

We next sought to determine which regions of SLAMR interact with the protein components of its interactome. Thus, we carried out RNAse protection assays to identify interacting (protected) regions within SLAMR. Following RNAse-A digestion and sucrose cushion centrifugation, (Fig. 8A)[59], we identified multiple RNAse-protected fragments from SLAMR, one of which showed a major peak spanning a 220nt segment in the middle of its sequence between 898-1130 nt (Fig. 8B). Additionally, two other fragments showed a low number of reads and/or were incomplete. We next asked whether the 220 nt fragment is sufficient to interact with either CamKIIα or Vimentin. We, therefore, prepared a biotin-labeled sense probe for the major peak (898-1130 nt probe), and another probe from a different region of SLAMR corresponding to a minor peak (92-289 nt probe) (Fig. 8C) and carried out pulldown experiments using lysates from mice hippocampus as input. In addition, we also included a SLAMR full-length sense probe as a positive control and an antisense probe as a negative control. Western blot analysis confirmed that the 898-1130 nt probe identified by RNAseq was sufficient to pulldown both proteins. CaMKIIα (*n* = 11-12; vs Antisense ****$p < 0.0001$, *$p < 0.05$; vs sense ##$p = 0.0032$, One-way ANOVA + Tukey's test; Fig. 8 D, E) and Vimentin (*n* = 5-6; **$p < 0.01$. One-way ANOVA + Tukey's test; Fig. 8 D–F) showed enrichment with the 898-1130 nt probe compared to the antisense

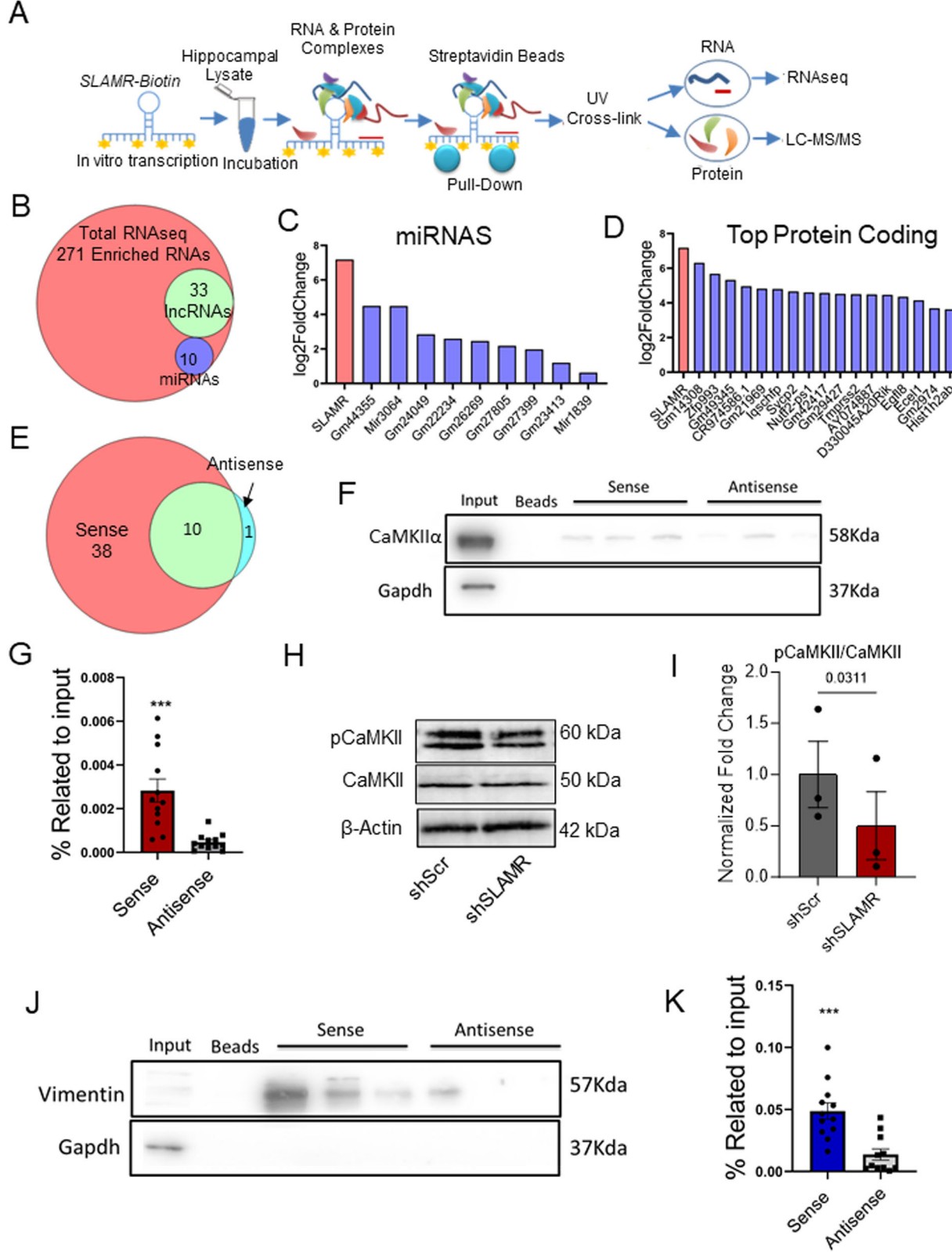

probe. These results indicate that the 898-1130 nt region of SLAMR is important for the formation of RNA-associated protein (RAP) complexes.

RAP complexes can mediate the interaction between lncRNAs and the motor proteins that transport them[60]. Thus, we sought to explore whether the 220 nt region (between 898 to 1130 nt) of SLAMR participates in SLAMR trafficking. For this experiment, we designed two new

MS2-SLAMR constructs; one with the CaMKIIα/Vimentin binding region deleted, MS2-SLAMRΔ898-1130, and one with the other fragment deleted, MS2-SLAMRΔ92-289, as control (Fig. 8G). We imaged them as previously described for MS2-SLAMR: tdGFP-MCP (Supplementary Fig. S6A–F, Supplementary Movies S6–7). Interestingly, the 92-289 nt deletion decreased both the anterograde and retrograde MS2-SLAMR velocity compared with the intact SLAMR, while the 898-1130 nt deletion

**Fig. 6 | lncRNA SLAMR interacts with several miRNAs and proteins.**
**A** Experimental design using pull-down purification assay with SLAMR full-length biotinylated probes. **B** Venn-diagram indicates the number of total RNAs enriched in the sense condition, number of miRNAs, and lncRNAs. **C** miRNAs identified after small RNA sequencing interacting with SLAMR. **D** Log2Fold change of the top 20 coding and non-coding genes after pulldown total RNAseq. **E** Venn diagram represents the number of proteins identified by LC-MS/MS for each experimental condition. **F** Representative CaMKIIα western blot. Three lanes of replicative pull-downs using sense and antisense SLAMR are shown. **G** Bar diagram of the results obtained after WB analysis of CaMKIIα based on the percentage of input. N = 12, 4 experiments x 3 replicates (***$p$ = 0.0002, two-tailed Unpaired t-test; data are presented as mean$n$ ± SEM). **H** Western blot analysis of β-actin, phosphorylated CaM-KII(p-CaMKII), and CaMKII isolated from primary hippocampal neurons transduced by either shScr or shSLAMR lentivirus synaptoneurosomes. **I** Quantification of (**H**) normalized to β-actin, relative to shScr. ($n$ = 3 biological replicates, two-tailed student's t-test; mean$n$ ± SEM). **J** Representative Vimentin immunoblot. Three lanes of replicative pulldowns using sense and antisense SLAMR are shown. **K** Bar diagram represents the results of WB analysis for Vimentin over the input percentage. N = 12, 4 independent pull-downs with 3 replicates each (***$p$ = 0.0003, two-tailed Unpaired t-test; mean$n$ ± SEM).

significantly increased the anterograde but not the retrograde velocity of MS2-SLAMR ($n$ = 9,11,9 neurons respectively MS2-SLAMR, MS2-SLAMRΔ92-289, MS2-SLAMRΔ898-1130; exact $p$-values shown; One-way ANOVA + Tukey's test, Fig. 8H, I, Supplementary Fig. S6A–F and Movies S6-7). Both the 92-289 nt and 898-1130 nt deletions significantly decreased the percentage of mobile MS2-SLAMR granules to 23.9 ± 3.6% and 24.3 ± 3.8%, respectively, compared with the intact SLAMR which had 46.9 ± 4.7% mobile granules, ($n$ = 11,9,10 for MS2-SLAMR, MS2-SLAMRΔ92-289, and MS2-SLAMRΔ898-1130, respectively; exact $p$-values shown; One-way ANOVA + Tukey's test; Fig. 8J). Furthermore, both these deletions decreased the percentage of MS2-SLAMR mobile granules within dendritic spines and the number of particles that changed directions at spines occupied with MS2-SLAMR from ~78% to ~69% for MS2-SLAMRΔ92-289 and ~55% of MS2-SLAMRΔ898-1130 RNA granules (Fig. 8K, L, Supplementary Fig. S6G–L, and Movies S8–9). Together, these results demonstrate that SLAMR integrity is critical for its transport within dendrites.

## SLAMR is required for consolidation, but not for acquisition, extinction, and recall of CFC

SLAMR's enrichment in the CA1 following CFC, its role in activity-dependent synaptic structural changes, its trafficking into spines, and its interactome support a key role for SLAMR in mediating Long Term Memory (LTM). Therefore, we assessed the effect of loss-of-function of SLAMR by RNAi-mediated knockdown using Gapmer oligonucleotides in the CA1 of mice in multiple memory processes such as acquisition, consolidation, extinction, and recall of contextual fear memory. We first designed three different Gapmer antisense oligonucleotides that were tested in primary mouse hippocampal neuronal cultures. These neuronal cultures were transfected independently with different Gapmers that either targeted SLAMR or a negative control (NC) sequence that does not have a specific target in mice (5' AACACGTC-TATACGC 3')[22]. Total RNA was isolated 42 and 72 hours after transfection (Supplementary Fig. S7A). The qRT-PCR analysis of these RNAs indicated that the levels of SLAMR 72 hours after transfection with Gapmer_1 were significantly reduced compared to the negative control (Gapmer1 = 0.286 ± 0.051, NC = 1 ± 0.150, $n$ = 3-4 per condition; ***$p$-value < 0.005, One-Way ANOVA + Dunnett's test; Supplementary Fig. S7A) and with higher efficiency than Gapmers 2 and 3. Furthermore, before performing the behavioral experiments, we tested Gapmer_1 (named SLAMR_Gapmer for the following experiments) in vivo to evaluate its efficiency. Mice received bilateral stereotaxic infusions in the dorsal CA1 hippocampal area of SLAMR_Gapmer using the JetSI delivery method. 72 hours after the surgery, mice were euthanized, and the area around the infusion was dissected from the hippocampus to isolate the RNA. The qRT-PCR results indicated that SLAMR_Gapmer was able to induce a 40% reduction in SLAMR expression in the hippocampus (NC = 1 ± 0.142, SLAMR_Gapmer = 0.630 ± 0.059, $n$ = 7 per condition; *$p$-value < 0.05, Student's t-test; Supplementary Fig. S7B).

Next, we evaluated the role of SLAMR in the early stages of memory acquisition. For this experiment, bilateral cannulas targeting the CA1 were implanted into mice. One week after the surgery, mice were divided into 3 groups that received a single infusion of their corresponding Gapmer into the CA1 dorsal area: SLAMR_Gapmer, NC_Gapmer, or Sham (non-infused). 72 hours after the infusion, all mice were trained in CFC, and 1 hour after the training, they were tested in the same context for 5 minutes without any shock delivery to evaluate the acquisition of conditioned fear response (Fig. 9A). The results indicated that silencing the expression of SLAMR in dorsal-CA1 did not affect the expression of fear during the training (Supplementary Fig. S7C–F) nor the acquisition of the conditioned response (SLAMR = 51.53 ± 5.55; Control=49.87 ± 5.27; Sham=47 ± 6.2; Fig. 9B, C). All groups (Sham, NC, and SLAMR_Gapmer) showed similar percentages of freezing values in both training and test sessions.

To evaluate the role of SLAMR in the consolidation of CFC memory, we designed the following experimental procedure similar to the previous one (Fig. 9D): 72 hours after the infusion, all mice were trained in CFC, and 24 hours after the training, they were tested in the same context for 5 minutes without any shock delivery to evaluate the memory consolidation. Results showed no differences between groups in the expression of fear during the training (Supplementary Fig. S7D–G). However, we found a significant reduction in the percentage of total freezing time during the 24-hour test in those mice that received the Gapmer silencing SLAMR expression. In contrast, the sham and negative controls showed similar values during the test session (SLAMR = 41.016 ± 3.579; Control=62.754 ± 7.385; Sham=58.895 ± 7.162; ##$p$ < 0.05, **$p$ < 0.01, Two-way ANOVA + Tukey's test; Fig. 9E and F). Together, these results indicate that the reduction in SLAMR expression in dorsal CA1 impaired the LTM consolidation of the CFC, while the expression of fear remained intact during the training.

Additionally, if SLAMR was knocked down 24 hours after training (Fig. 9G, Supplementary Fig. S7E–H) when the memory had already consolidated, this did not interfere with memory expression (Fig. 9H). Also, extinction training performed 72 hours after SLAMR inactivation did not show any differences in the percentage of freezing values between groups (Fig. 9I). Furthermore, the recall test performed 24 hours after extinction training, verified the extinction efficiency by the high reduction in the percentage of freezing levels in all the groups. Recall test values are similar between treatments indicating that SLAMR activation in dorsal CA1 is not necessary for this process (SLAMR = 21.53 ± 3.04; Control=19.60 ± 3.26; Sham=22.55 ± 4.08; Fig. 9J, K).

On the other hand, previous studies demonstrate that activity-dependent transcriptional changes induced by CFC training take place not only in CA1 but also in the CA3 hippocampal subregion[61]. To verify this observation, we performed a behavioral study silencing the expression of SLAMR in the dorsal CA3 of the hippocampus following the same procedure for the CA1 experiments (Fig. 9L; Supplementary Fig. S8A). In this case, data showed no differences between any of these groups in the fear expression during the training (Fig. 9M), as well as in the percentage of freezing time during the test for memory consolidation (Fig. 9N, O). All the groups included in this study yielded similar values. These results indicate that SLAMR is not necessary for memory consolidation in the dorsal CA3 hippocampal subregion. Taken together, these results indicate that SLAMR is critical for memory consolidation, whereas the expression of fear, acquisition, extinction, and recall functions remain intact upon reduced SLAMR levels in dorsal CA1 and CA3 hippocampal areas.

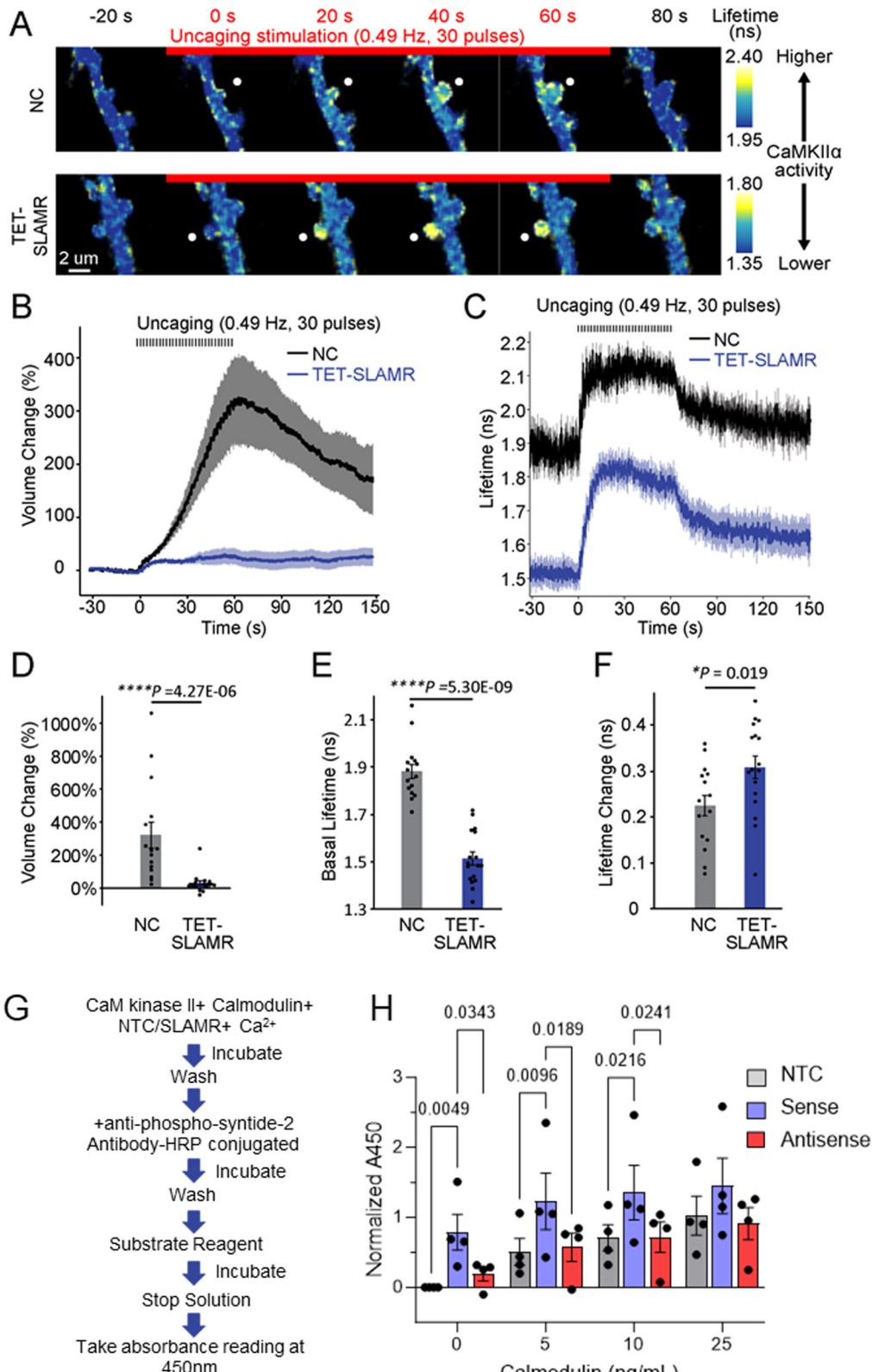

**Fig. 7 | SLAMR regulates transient structural spine plasticity and basal CaM-KIIα activity in CA1 pyramidal neurons. A** Timelapse fluorescence lifetime imaging of a CaMKIIα FLIM-FRET sensor in dendrites of CA1 pyramidal neurons co-expressing SLAMR shRNA-mCherry (TET-SLAMR) or scrambled shRNA-mCherry (NC-mCherry). CaMKIIα is locally activated by a single spine glutamate uncaging (30 pulses, 0.5 Hz). Scale bar = 2 μm. **B**, **C** Mean time courses of glutamate uncaging-induced volume change (**B**) and fluorescence lifetime of a CaMKIIα FLIM-FRET sensor (**C**) in stimulated dendritic spines expressing NC-mCherry (*n* = 15 spines/5 neurons) or TET-SLAMR (*n* = 17 spines/5 neurons). Data are presented as mean *n* ±

SEM. **D**–**F** Quantifications of volume change (**D**), basal lifetime of CaMKIIα before glutamate uncaging (**E**), and glutamate uncaging-induced lifetime change of CaMKIIα sensor (**F**) (NC-mCherry *n* = 15 spines/5 neurons, TET-SLAMR *n* = 17 spines/5 neurons; Two-tailed Mann-Whitney U-test (**D**–**E**) Two-tailed unpaired Student's t-test (**F**); the data are presented as mean ± SEM). **G** Schematic of CaMKII in vitro activation assay. **H** CaMKII activation quantified by Normalized A450 Absorbance values. Normalized by blank subtraction (0 ng/mL Calmodulin, No Template Control (NTC) absorbance value), (N = 4 biological replicates per group, Two-way ANOVA + Tukey's test; data are presented as mean *n* ± SEM).

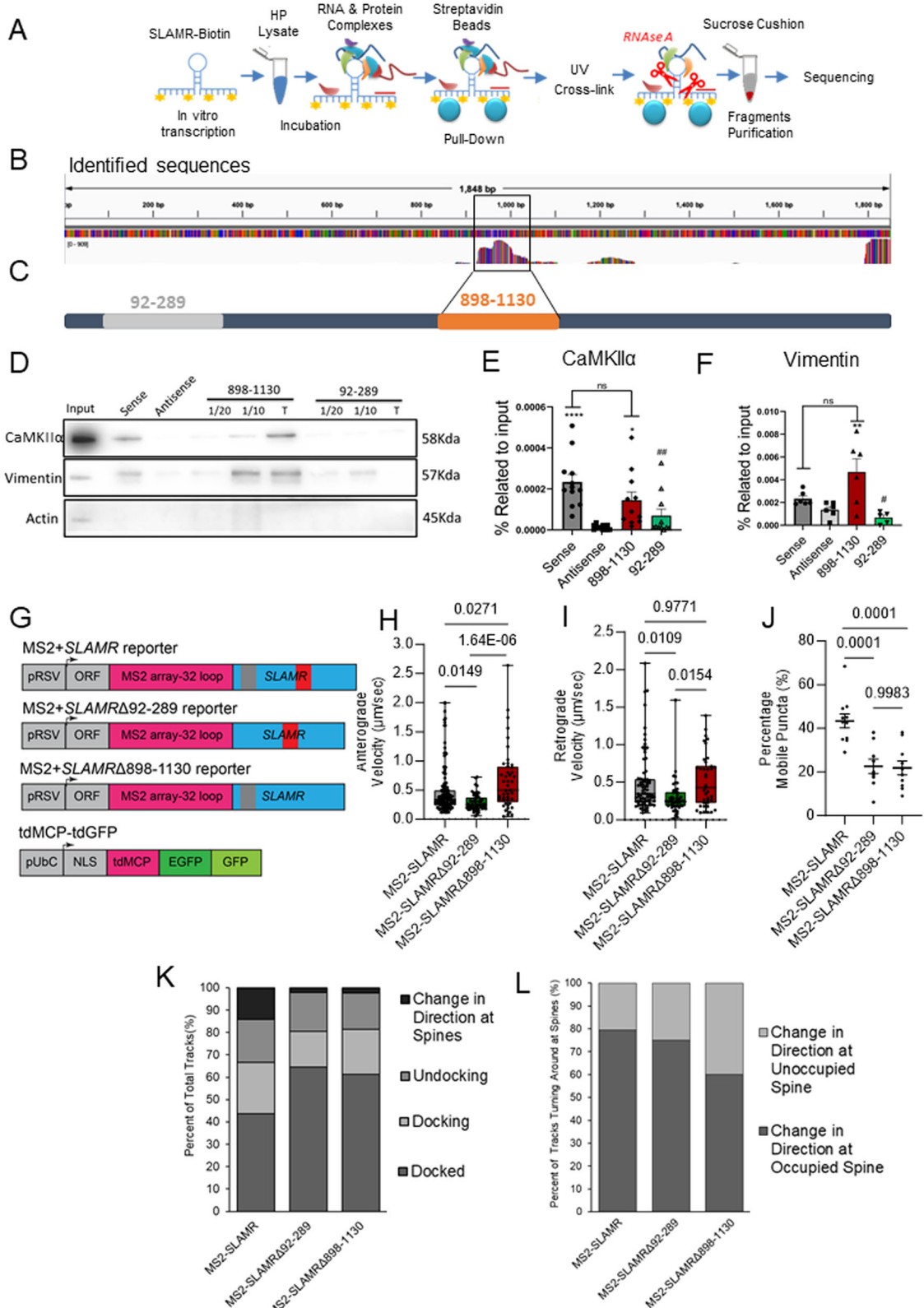

## SLAMR expression in dorsal CA1 is not required for spatial learning and memory

It is well known that the dorsal CA1 of the hippocampus plays an essential role in spatial learning and memory processes[62,63]. To determine whether SLAMR is implicated in other hippocampal-dependent behaviors, we performed the Morris Water Maze (MWM) assay to evaluate spatial hippocampal-dependent functions. Mice were divided into three different groups and trained in the MWM test for a week or a single day or just exposed to a session of swimming without a platform to discard any influence induced by the novel environment or the exercise (Fig. 10A and B). Exploratory behavior was similar between the 1 week training group, 1-day training group, and the swimming session group (Fig. 10C). There were no differences at the basal level between different experimental groups. 1 hour after finishing their respective

**Fig. 8 | CaMKIIα and Vimentin interact with a specific fragment of the lncRNA SLAMR. A** Schematic of approach to identify protein-protected fragments of SLAMR using a pull-down strategy. **B** Mapping of seq-Reads of the mouse genome using Salmon and IGV software. **C** Scheme of cloned SLAMR fragments used to design specific probes. **D** Representative images of CaMKIIα, Vimentin, and Actin from WB with PD samples. **E–F.** Plots of WB quantification for CaMKIIα (Independent experiments, N = 11-12, vs Antisense ****$p$ = 4.04E-05, *$p$ = 0.0229; vs sense ##$p$ = 0.0032. One-way ANOVA + Tukey's test; data as mean ± SEM) and Vimentin (Independent experiments, N = 5-6, One-way ANOVA + Tukey's test; **$p$ = 0.008, #$p$ = 0.002; data as mean ± SEM). **G** Schematic of MS2-SLAMR truncated constructs. **H–J** Quantifications of Supplementary Fig. 7A–F Kymographs. **H** Deletion of the alternative (92-289) fragment decreased the anterograde speed while deletion of Vimentin/CaMKII (898-1130) binding site increased the speed of MS2-SLAMR:MCP granules compared to the full-length MS2-SLAMR (MS2-SLAMR $n$ = 134 tracks/11 neurons, Δ92-289 $n$ = 56 tracks/9 neurons, Δ898-1130 $n$ = 45 tracks/10 neurons;

One-way ANOVA + Tukey's test; data are presented as min/max). **I** Deletion of the alternative (92-289) fragment decreased the retrograde speed while deletion of the vimentin/CaMKII (898-1130) binding site led to no change compared to the full-length MS2-SLAMR (MS2-SLAMR $n$ = 80 tracks/11 neurons, Δ92-289 $n$ = 49 tracks/9 neurons, Δ898-1130 $n$ = 39 tracks/10 neurons; One-way ANOVA +Tukey's test; data are presented as min/max). **J** Deletion of both the control (92-289) and the vimentin/CaMKII (898-1130) binding site reduced the percent mobile MS2-SLAMR:MCP granules compared to the full-length MS2-SLAMR (MS2-SLAMR $n$ = 11 neurons, Δ92-289 $n$ = 9 neurons, Δ898-1130 $n$ = 10 neurons; One-way ANOVA + Tukey's test; data as mean ± SEM). **K, L** Quantifications of Supplementary Fig. 7D–F Kymographs. **K** Distribution of how MS2-SLAMR, MS2-SLAMRΔ92-289, and MS2-SLAMRΔ898-1130 interact with dendritic spines. **L** Both the deletions of the alternative fragment and vimentin/CaMKII binding sites decrease the percentage of times that MS2-SLAMR turned around at dendritic spines already occupied by MS2-SLAMR.

trainings, mice were euthanized, and the brains were isolated to dissect the CA1 dorsal area by LCM. The RNA obtained was reverse transcribed and qRT-PCR was performed to measure SLAMR expression. The qRT-PCR experiments did not show any significant differences between groups in dorsal CA1 in any of the three different conditions, indicating that SLAMR expression is not increased in this hippocampal subregion by MWM training (1 week=1.07 ± 0.13; 1 day=1.34 ± 0.19; Swimming=1 ± 0.15; Fig. 10D).

To confirm this observation, a second experiment was carried out manipulating the expression of SLAMR in dorsal CA1 following a similar procedure from previous experiments of this study. As the experimental design indicates (Fig. 10E), mice were cannulated to target dorsal CA1 (Supplementary Fig. S8B). A week after the surgery, mice were treated with SLAMR_Gapmer, NC_Gapmer, or Sham (non-infused). 72 hours after the infusion, mice were trained in the MWM for 7 days, with 4 trials each from randomly assigned starting positions. Then, 24 hours after finishing the training, a long-term memory test was performed to evaluate the memory consolidation by removing the platform and using a novel starting position. During the training, none of the groups showed significant changes between them in latency, distance, or velocity (Fig. 10F). These results indicate that spatial learning, motor responses, and/or motivation are all intact after cannula implantation and genetic manipulation in this brain region. Similarly, mice spent similar amounts of time in the target quadrant and number of crossings over the previous platform position (Fig. 10G–I), revealing that SLAMR reduction in dorsal CA1 does not affect spatial memory consolidation in this behavioral paradigm.

## Discussion

Previous studies have pointed to the significance of long noncoding RNAs (lncRNAs) as key components in transcriptional changes relevant to learning and LTM[12,13,22,27,28,64,65]. However, our understanding of lncRNAs specifically localized within dendrites and their precise mechanisms of action within the synapse is still limited. Here we describe the identification, regulation, mechanism, and function of a previously unreported lncRNA, which we have named SLAMR, within dendrites of hippocampal neurons.

### SLAMR functions as a "master" regulator of structural plasticity

Dendritic arborization and the ability to regulate structural and synaptic plasticity are critical for neurons to function in learning and LTM. Studies from our lab and others reveal that lncRNAs have important and diverse roles in regulating dendritic arborization, spine density, and plasticity. Loss-of-function of some lncRNAs leads to a reduction in dendritic arborization, spine density, and morphology[22] while others have no effect on arborization, but exhibit strong reductions of spine density, morphology, and synaptic plasticity[13]. Still, loss-of-function of other lncRNAs can also lead to increased spine density[12,66] and size yet decreased dendritic arborization and structural

plasticity[66]. Importantly, glutamate uncaging followed by two-photon imaging of dendritic spines, shows that ADEPTR-deficient spines fail to undergo transient changes in spine morphology[13]. Interestingly, in this study, we find that loss-of-function of SLAMR produced a significant decrease in dendritic arborization, although spine density and morphology of dendritic spines were not altered. However, single spine stimulation and timelapse imaging of spine morphology suggest that the spines are also deficient in undergoing activity-dependent structural changes. Taken together, these results suggest that key features of neuronal architecture could be modulated by the expression of SLAMR.

While the loss of certain long noncoding RNAs (lncRNAs) has been linked to deficits in neuronal morphology, the consequences of lncRNA gain-of-function remain less explored. In our study, we observed that the overexpression of SLAMR-alone resulted in enhanced dendritic arborization, spine density, and morphology. Furthermore, we found that the silencing of SLAMR led to a reduction in global translation, likely by decreasing the expression of key translation regulators, such as eIF2α and S6 Kinase.

Previously, we demonstrated that silencing the molecular motor KIF5C led to decreased translation in synaptoneurosomes, while overexpression of KIF5C resulted in overall enhancement in translation[8]. This result prompted us to investigate the potential relationship between the expression of KIF5C and SLAMR. Our findings indicated that the loss of KIF5C function led to reduced levels of SLAMR in synaptoneurosomes, whereas KIF5C overexpression increased SLAMR abundance. Conversely, the loss of SLAMR function resulted in decreased levels of KIF5C, and SLAMR overexpression led to an increase in KIF5C levels. This intriguing reciprocal regulation suggests that the overexpression of SLAMR activates the expression of KIF5C, likely explaining the observed increase in translational and morphological changes. Our results imply that the regulation of dendritic transport by SLAMR relies on the motor protein KIF5C, and KIF5C functions as the carrier for SLAMR in dendrites. An increase in SLAMR levels in the neuron may increase the demand for KIF5C expression to facilitate efficient transport. Conversely, a decrease in KIF5C availability leads to a reduction in SLAMR availability. Therefore, the interaction between these two components, cargo and motor, likely underlies the observed reciprocity.

### SLAMR is transported into dendritic spines through the molecular motor KIF5C

While dendritic localization of populations of mRNAs and a few lncRNAs have been described[13,47,67–72], the mechanism by which these RNAs are transported is much less understood. Several studies have demonstrated that kinesins play a crucial role in RNA transport[8,70,73,74]. Nonetheless, while there are hundreds of RNAs shown to be localized to hippocampal dendrites, we do not know which kinesins mediate their transport and how they do so.

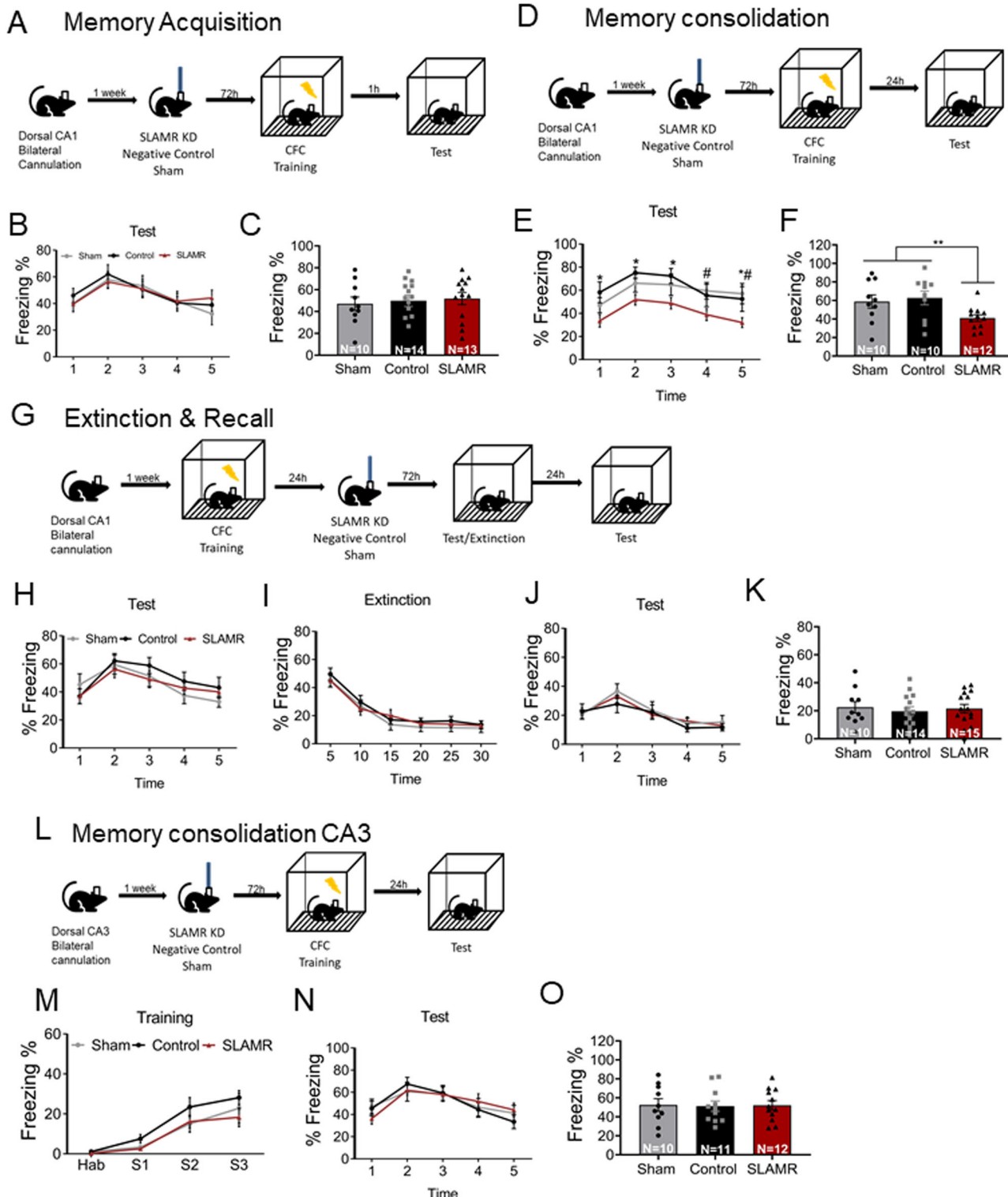

Recently, our research demonstrated that KIF5C is associated with approximately 700 RNAs, indicating its likely role in mediating their dendritic localization, thereby making it a promising candidate for lncRNA transport. Furthermore, in our earlier investigation of the dendritically localized ADEPTR lncRNA, we discovered that its localization is dependent on KIF2A, a kinesin previously not associated with RNA localization[13]. Following a similar approach, we found that KIF5C is a key mediator of SLAMR transport to dendrites. Collectively, our findings indicate that kinesins play a significant role in facilitating the dendritic targeting of lncRNAs.

To gain a comprehensive understanding of the dendritic transport of SLAMR, we conducted a quantitative analysis by expressing SLAMR tagged with MCP and MS2 in hippocampal neurons. This approach not only provided insights into the transport kinetics mediated by KIFs but also revealed that SLAMR could access various types of dendritic spines. Notably, SLAMR exhibited a preference for spines enriched with PSD95, indicating a particular affinity for mature spines. Our in-depth analysis of SLAMR transport unveiled a range of complex dynamics, including docking, undocking, and alterations in transport direction at mature spines. Particularly intriguing was the

**Fig. 9 | SLAMR is implicated in contextual fear memory consolidation. A** For the acquisition experiment, mice were implanted with bilateral cannulas in the dorsal CA1 and then divided into three groups (sham, NC, and SLAMR_Gapmer) and infused with their corresponding agent 72hrs before the CFC training. **B, C** 1 hr after training these mice were placed again in the same context for a 5 min test. There were no differences between groups in the early retention test. (Sham $n = 10$, Control $n = 14$, SLAMR $n = 13$; data are presented as mean ± SEM). **D** SLAMR was KD in dorsal CA1 of mice implanted with bilateral cannulas 72hrs before the training and tested 24 hrs later in the same context. **E, F** The long-term memory test was carried-out 24h after the training and shows a significant reduction in the percentage of freezing time in SLAMR-KD compared to controls (Sham $n = 10$, Control $n = 10$, SLAMR $n = 12$; (E) Two-way ANOVA + Uncorrected Fisher's LSD (*control vs SLAMR: min 1 $p = 0.034$, min 2 $p = 0.0195$, min 3 $p = 0,0183$, min 5 $p = 0.0401$; #Sham vs SLAMR: min 4 $p = 0.0391$, min 5 $p = 0.0125$) and (F) Two-way ANOVA +

Tukey's test; control vs. SLAMR **$p = 6.88E-06$, sham vs. SLAMR **$p = 2.47E-04$) Data as MEAN ± SEM. **G** Experimental design where SLAMR was KD after memory consolidation. **H** SLAMR KD 24hrs after finishing the training does not affect the expression of fear during the long-term memory test (Sham $n = 10$, Control $n = 14$, SLAMR $n = 15$; data as mean ± SEM). **I** The percentage of freezing during the 30 mins of the extinction process does not show any differences between groups (Sham $n = 10$, Control $n = 14$, SLAMR $n = 15$; data as mean ± SEM). **J, K** The Recall test was performed 24 hrs after the extinction shows an effective reduction in the expression of freezing in all the groups (Sham $n = 10$, Control $n = 14$, SLAMR $n = 15$; data as mean ± SEM). **L** Experimental design for SLAMR manipulation in CA3. **M** Plots show a similar percentage of freezing expression between groups during the training as well as (**N**), (**O**) 24h after the training the long-term memory test was performed (Sham $n = 10$, Control $n = 11$, SLAMR $n = 12$; data as mean ± SEM).

observation that SLAMR often changed direction when encountering spines already occupied by another SLAMR. This directional shift diminished when critical protein-binding regions, such as the CaMKII binding region, were removed from SLAMR. This result implies that neurons possess mechanisms for detecting and regulating SLAMR abundance within spines, and the presence of protein-binding regions plays a crucial role in this regulation. While the precise physiological implications of these transport characteristics remain somewhat unclear, our observations highlight the dynamic nature of lncRNA localization within spines, suggesting that lncRNAs can reside within spines and that their localization is subject to dynamic changes.

The MS2:MCP system enabled us to investigate the behavior of SLAMR in response to spine stimulation. We conducted two-photon glutamate uncaging and observed the dynamics of MS2-tagged SLAMR within dendritic spines, categorizing them into those undergoing structural plasticity and those remaining unchanged. Interestingly, spines exhibiting structural plasticity demonstrated an increased recruitment of SLAMR within the neighborhood of the stimulated spine approximately 5 minutes after stimulation. It is important to note that while this transport is likely facilitated by a motor protein, it is not clear if it is KIF5C-dependent and will have to be investigated in future studies. These results support that SLAMR is necessary for structural plasticity and that stimulation increases the local abundance of SLAMR.

### SLAMR expression constrains CaMKIIα activity
SLAMR interactome characterization identified several coding and noncoding RNAs, and proteins, collectively forming a multi-protein-RNA complex. Notably, among these interactors, we identified CaMKIIα as an interacting protein. Given the well-established roles of CaMKIIα in synaptic plasticity and memory[75], we conducted several experiments to validate its interaction with SLAMR and assess its significance.

Independent experiments not only confirmed the interaction between SLAMR and CaMKIIα but also pinpointed a specific 220 nucleotide fragment that is sufficient for this interaction. Intriguingly, the deletion of this fragment also affected the dendritic transport of SLAMR. Considering the relative abundance (with CaMKIIα being highly expressed compared to SLAMR), we speculated that SLAMR might play a role in mediating the activity of CaMKIIα in dendrites. Previous studies demonstrated that CaMKIIα is localized in dendrites and translocated to spines in response to neuronal activity[76–78]. Consistent with a role in modulating CaMKII activity, we observed that the loss of SLAMR function resulted in a reduction in the active pool of CaMKIIα (T282 phosphorylated) in synaptoneurosomes. Furthermore, in organotypic cultures, a CaMKII activity sensor revealed a baseline decrease in CaMKII activity when SLAMR expression was reduced. Additionally, upon spine stimulation, there was an overall reduction in CaMKII activity, with the peak activity during stimulation barely reaching the prestimulation CaMKII activity in the controls.

Nevertheless, CaMKII activity still significantly increased upon spine stimulation in SLAMR KD. This increase may be due to several reasons, including residual SLAMR, or a SLAMR independent mechanism. Additionally, our in vitro assay revealed that SLAMR can significantly enhance CaMKII activity. Taken together, these results suggest that SLAMR is a previously undescribed modulator of CaMKIIα function in hippocampal neurons.

### SLAMR exhibits region-specific and memory-specific functions
It is well-established that the dorsal hippocampus plays a pivotal role in various forms of associative and spatial learning and memory functions[30,62,63,79–82]. However, our investigation revealed that silencing SLAMR in the dorsal CA1 region did not impede spatial navigation learning and its subsequent long-term consolidation in the Morris water maze (MWM). This result suggests that SLAMR's specific function may be tied to precise forms of neuronal activation (e.g., fear conditioning versus place-based navigation) within distinct CA1 neuron subtypes characterized by unique properties[83]. A similar reasoning may be applied to our study of the CA3 hippocampal region[83]. Prior research has indicated that while the CA1 region is essential for both the acquisition and consolidation of memory, the CA3 region is primarily associated with rapid acquisition responses[84–87]. Consistent with these findings, our results show that SLAMR was enriched in CA1 but not in CA3 during the early stages of memory acquisition. Additionally, SLAMR RNAi in CA3 did not impair contextual fear conditioning memory consolidation.

The intriguing memory-specific role of SLAMR raises questions about its precise mechanisms. Although SLAMR and its interacting partners are likely expressed in various cell types, manipulating SLAMR in the dorsal CA1 region specifically impairs the consolidation of contextual fear memory. This remarkable precision suggests that SLAMR is involved in a specific cellular process within components of the fear circuitry, primarily influencing the consolidation of contextual fear memories. Given that this is the first study to provide a detailed analysis of a lncRNA's role in different types of memory and various memory phases, there is currently no evidence to suggest similar roles in other lncRNAs. For instance, in a study looking at transcript profiles change as a function of sleep versus Sleep deprivation prior to CFC). Delorme et al. [39] reported significant changes in lncRNA D17RIK (that we now term SLAMR) expression in the hippocampus of mice after sleep deprivation after contextual fear conditioning (CFC). When we examine prior findings from other research groups that employed non-constitutive silencing techniques, such as Antisense Oligonucleotides (ASOs), in specific brain areas, we observe patterns that suggest a similar trend. For example, Gomafu and Adram are two lncRNAs that are well expressed in different cell populations in the brain, particularly in the prefrontal cortex (PFC). However, silencing Adram before fear conditioning training did not affect fear acquisition but impaired the extinction process[88]. Conversely, silencing Gomafu before training impaired fear acquisition while memory consolidation remained

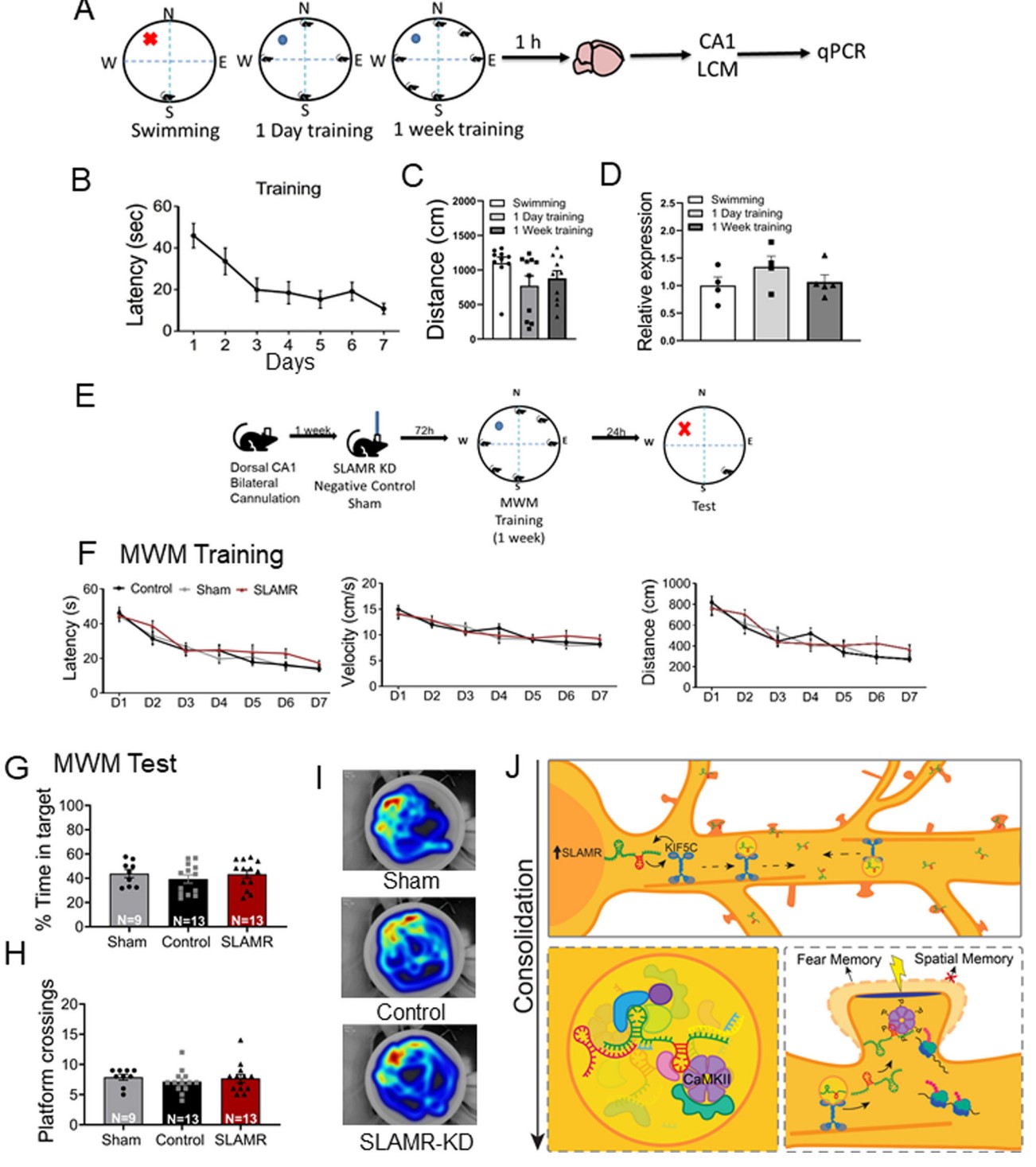

**Fig. 10 | SLAMR is not enriched in dorsal CA1 after MWM training, nor is it required for spatial learning and memory. A** Experimental design. **B** Latency of mice in MWM during 1 week of training ($n = 10$ mice; data are presented as mean ± SEM). **C** Distance during the first trial or a single swimming session ($n = 10$ mice per group; data as mean ± SEM). **D** RT-qPCR results show that lncRNA SLAMR is not increased in dorsal CA1 1 hr after finishing a single session of swimming exercise, 1 day, or 1 week of training in MWM ($n = 4$ mice per group; data as mean ± SEM). **E** Schematic representation of SLAMR genetic manipulation for MWM study. **F** Latency values did not show any differences between groups during the learning and the long-term memory test. Also, no differences in distance and velocity values indicated behavioral changes induced by the genetic manipulation of SLAMR. (Latency: Sham $n = 9$, Control $n = 13$, SLAMR $n = 13$, Velocity and Distance: Sham $n = 9$, Control $n = 13$, SLAMR $n = 12$; data as mean ± SEM). **G**, **H** Test

results in percentage of time spent in target quadrant and platform crossings show no differences between groups for consolidation of spatial memory. Data as mean ± SEM. **I** Representative heat maps indicated similar exploratory behavior during the test session for all three conditions. **J.** Model of SLAMR during consolidation. Top panel: Learning increases SLAMR expression, which reciprocally regulates KIF5C expression. KIF5C transports and deposits SLAMR throughout the dendrite and within dendritic spines. Bottom panel left: Model of a vesicle containing SLAMR and identified interactor CaMKII, along with numerous other proteins, lncRNAs, microRNAs, and mRNAs as suggested by the pulldown experiments. Red hairpin indicates binding region to CaMKII/Vimentin. Bottom panel right: Stimulated dendritic spine showing structural plasticity, the active recruitment of SLAMR, phosphorylation of CaMKII, and increased local protein synthesis. SLAMR participates in this role in fear memory but not spatial memory.

intact[89]. Taken together lncRNAs may indeed have specific roles in distinct groups of cells within the same brain structure, each activated by inputs from diverse brain areas.

In summary, our research has unveiled a previously undescribed long non-coding RNA (lncRNA) that we have named SLAMR. We have shown that this lncRNA is induced by specific experiences in the dorsal CA1 region and is transported along dendrites through KIF5C (Fig. 10J). Our findings shed light on the experience-dependent, cell-specific regulation of lncRNA expression and the intricate mechanisms by which lncRNAs can influence crucial cellular processes, such as protein synthesis and the function of key signaling molecules like CaMKIIα. Moreover, our results emphasize the importance of understanding the many yet unknown functions of lncRNAs within the nervous system and highlight the need to unravel their roles in concert with coding RNAs and proteins in collectively mediating cellular functions.

## Methods
Our research complies with all relevant ethical regulations. Housing and experimental procedures performed with mice and rats were approved and supervised by the Institutional Animal Care and Use Committee of the Herbert Wertheim UF Scripps Institute for Biomedical Innovation & Technology and Max Planck Florida Institute for Neuroscience.

### Animals
For this study, we used C57BL6 adult male mice 8-10 weeks old provided by Jackson Laboratories. These mice were housed at the Wertheim UF Scripps Institute in groups of 5 and maintained on a 12 hr light/dark cycle with *ad libitum* access to water and food and kept at an ambient temperature of 22 °C and 28 – 72 % humidity Timed pregnancy Sprague Dawley rats and C57BL6 mice were kept at the Max Planck Florida Institute for Neuroscience and maintained on a 12 hr light/dark cycle with *ad libitum* access to water and food at an ambient temperature of 22 °C and 35 – 60 % humidity. In vivo experiments were carried out during the light part of the cycle light/dark cycle. All in vitro experiments were performed in primary hippocampal cell cultures obtained from CD1 mouse pups, except for MS2:MCP imaging experiments which were performed in primary hippocampal cell cultures obtained from Sprague dawley rat pups. Rat neurons were used for these experiments as they have the ideal structure for observing the movement of these big complexes formed by the RNA with the MS2 loops (32) and the MCP molecules attached. They are also more resistant to triple transfections with large plasmids and extended live-cell imaging sessions.

CD1 pregnant females were purchased from Charles River and Sprague Dawley pregnant females were purchased from Jackson Laboratories.

### Statistical analysis
Levels of significance in this study are based on *p*-values calculated by GraphPad Prism 8 or 9 (Graph Pad Software) and derived using Student's t-test, Mann-Whitney U test, One-way or Two-way ANOVA. Tukey, Dunnett, or Sidak tests were used for post hoc analyses. Significance was defined as < 0.05. The details for each experiment as well as the number of replicates and statistical specifications are indicated in the figure legends, results, and Supplementary Data.

### The following methods can be found in more detail in the Supplementary Information
*RNAseq*
> *Interactome analysis*
> *Quantitative real-time PCR (qRT-PCR)*
> *Localization studies of SLAMR*
> *Primary Hippocampal Cell Cultures*
> *Loss of function analysis using Gapmers*
> *Constructs & Transfections*
> *Lentiviral production*
> *SLAMR-MS2: MCP transport timelapse video microscopy*
> *MS2-SLAMR two-photon glutamate uncaging experiments*
> *Synaptic protein extraction*
> *Morphology Assessments.*
> *Two-photon fluorescence microscopy and two-photon glutamate uncaging for analysis of spine morphology*
> *Puromycin Labeling*
> *RNA pull-down assay: Proteomic, transcriptomic, and protected fragments analysis*
> *Immunocytochemistry (ICC)*
> *Organotypic hippocampal slice cultures and transfection*
> *Two-photon fluorescence microscopy and two-photon glutamate uncaging for analyses of spine morphology and CaMKIIα activity*
> *2-photon fluorescence lifetime imaging analysis*
> *CaMKII Activation Assay*
> *Behavior*

## Reporting summary
Further information on research design is available in the Nature Portfolio Reporting Summary linked to this article.

## Data availability
RNAseq data related to Figs. 1 and 6 were deposited to NCBI Gene Expression Omnibus with the accession numbers GSE214838 and GSE214839 respectively. Also, LC-MS/MS data included in Fig. 6 was deposited to MassIVE with the project number MSV000091477. Source data are provided with this paper. All other relevant data, for which there are no databases, are available from the authors upon request. Source data are provided with this paper.

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

## Acknowledgements

This work was supported by NIH grants (5R01MH094607-05, 1R21DA039417-01A1, 1F32MH131420-01A1, and 1R01MH119541-01A1). VR is funded by the Max Planck Society. MAK is supported by the DFG (Ki 502/9-1, 506658941). We sincerely thank Dr. Supriya Swarnkar for help with optimizing transfections, Ms. Kerriann Badal for behavioral experiments, Ms. Sabrina Perez for preparing all rat primary hippocampal neuron cultures, Steffan O'Korafor for helping with RIP optimizations, Ms. Xiaodan Kathy Liu for her technical support regarding the CaMKII sensor studies, Dr. Haruhiko Bito for the RCaMP1.07 plasmid, and the UF Scripps Biomedical Research Genomics, Proteomics, and Bioinformatics cores for RNAseq and proteomic analysis.

## Author contributions

I.E., J.L.W., and S.P. conceived of and designed experiments. I.E., J.L.W., Y.N., K.C., E.G., I.G., K.E.B., and B.R. conducted the experiments and analyzed the data. I.E., J.L.W., and S.P. wrote the manuscript. R.Y. provided expertise regarding single spine stimulation and the ex-vivo CaMKII activation assay. V.R. provided her expertise on single spine stimulation and RNA transport imaging and analysis. M.A.K. and V.R. contributed suggestions and revised the manuscript. All authors read and commented on the manuscript.

## Competing interests

The authors declare no competing interests.
