## [Peer Review File · Nature Communications]

Synaptically-targeted long non-coding RNA SLAMR promotes structural plasticity by increasing translation and CaMKII activityREVIEWER COMMENTS

Reviewer #1 (Remarks to the Author):

The authors of the study entitled "Synaptically recruited lncRNA SLAMR facilitates structural plasticity in fear memory consolidation by modulating translation" use behavioral training models and compare the differential gene-expression changes before and after training. This leads to the identification of SLAMR (RikD17) that is unregulated and localized to hippocampal cytoplasm - relative to nuclear. The authors then use Fish in primary hippocampal neurons and find punctate staining at dendrites. The authors then demonstrate that the mouse lncRNA translocates in the RAT neurons. It is a bit surprising that there is no discussion of isolating the rat version of this lncRNA before using mouse in an orthotopic manner. The lncRNA locus is syntenic in RAT (via a quick blast of mouse D17) therefore would seem better suited for a homotopic analyses rather than heterotrophic. Nonetheless the authors find that there is a lot of movement of RNA granules after training than before. However, these don't seem to be specific to dendrites and rather movement in all directions. Thus the thread is a bit confusing of a dendrite localized RNA in mouse, testing its movement in rat and finding that granules are moving in all directions. The authors further test a couple of proteins by knockdown and find KIF5C seems to mediate mouse SLAMR localization in rat neurons. The authors further show that loss of SLAMR decreases KIF5C and over-expression increased KIF5C. The authors also show that loss and gain of SLAMR results in less or more arborization in neurons, respectively. The authors then pull down SLAMR, unclear if in mouse or rat neurons, and find associated RNAs and proteins. They then further parse this down to a 200nt fragment that binds CaMKIIa and Vimentin. They then determine if the 200nt mouse region affects localization in rat neurons -- removing the 200nt binding regions results in less mobility. Finally the authors test the role of SLAMR in memory and recall etc. They used Gapmers to deplete what I think is mouse SLAMR in mouse neurons. They find SLAMR important for memory consolidation, but not other aspects tested. The authors further find that SLAMR is not relevant for other hippocampal functions.

Overall there is a lot of data in this study. However, the function of SLAMR remains unclear. There are characterizations of the mouse SLAMR in rat neurons. This is where most of the "function" is described. It is less clear that SLAMR has a clear function in mouse where I believe the original DEG were identified. There are so many quilted parts of data is very hard to determine how things fit together and or what SLAMR does in the context of training where it was discovered. I found that the mouse SLAMR-ms2 studies were not very convincing in relationship to the training where the authors identified SLAMR. Do all RNAs move more during training? If so then SLAMR would just be one of many mobile RNAs during this process.

I am concerned that a general reader would not follow the experiments nor how to continue to study SLAMR in the context of learning -- other than generating a mouse knockout and/or GOF model. I feel this study would be best suited in a learning neuroscience focused journal -- rather than the large and general readership of nature communications.

1) Fold changes and Pvalues should be listed for the reader to understand the effect sizes being measured. For example:

RNAseq data showed a significant reduction of Sox9 in the dorsal CA1 after CFC training exclusively in the C+S condition compared to both control groups. How much and how significant were the changes - should be listed in main text for any claim of "significance"

2) There are no statistics or quantification of "dendritic" localization in Figure 1J. This should be quantified and determined if significant before training as well.

3) "We used an shRNA plasmid to

471 constitutively silence the expression of SLAMR and alternatively, another one under a

472 doxycycline-inducible promoter (TET) for temporal control of SLAMR KD. A plasmid

473 expressing a scrambled sequence was used as negative control (NC). All plasmids also

474 express eGFP to visualize transfected neurons and assess morphological changes

475 (Figure 4A,B)."

It is very confusing in many places in the manuscript if the neurons are RAT based or Mouse, considering mouse SLAMR is being tested in rat. For example is the shRNA to an ectopic ally expressed mouse SLAMR in rat neurons? Also what is the effect size of SLAMR on "Arborization" of neurons in the experiments examined in text pasted above? For example Figure 4 H-K show only very small effects (10% to 15% in K etc...

4) The "movement" and Arborization studies are very confusing in RAT neurons. I am not sure how any of these findings relate back to how the RNA was discovered. I would suggest testing behavioral studies in a knockout model. Are the learning differences noticeable? The knockdown studies are not very convincing and most are found to be negative. It is also hard to understand how longer term responses could be affected by acute timeframes of knockdown (48-72 hrs).

Other major considerations:

A) Figure 1, Lines 173-180, 1017-1036, 1526-1544. The authors fail to mention how many animals were sacrificed for each condition i.e., sample size (n=) for the RNAseq.

B) MS2-MCP experiments in Figure 2, 3B-G, 7D, and on. The authors studied the activity and protein binding partners of murine lncRNA in the context of rat neuronal hippocampal cultures, in other words, they studied the function and RNP structure of a lncRNA in a different species without initial or similar experiments in the original species. While some lncRNA may act similarly in different species, the seminal experiments on a lncRNA's function should be mainly performed in the species of its origin, in this case, mouse. The authors found possible homologues in human and zebrafish but found none in rat. We question the validity and conclusions of these experiments considering this and whether function in a different species is significant and relevant, without first studying it in the context of its original species, mouse. The rat cell experiments must be repeated in mouse cells. We also stress the importance of indicating the cell species in their figures and figure legends.

C) The authors use two different species of primary hippocampal cells in their manuscript, however there are several examples of noting the cells, but not the species. The authors must clearly indicate the species of primary hippocampal cells throughout the manuscript (e.g., Lines 378-379, Figure 3A, 420-421, 442, Figure 3H-K, 444, 524, 545, 554, Figure 4A, Figure 5A, etc).

D) Figures 6F, 6J, and 6K. The authors identified protein partners of SLAMR via mass spec and sought to validate using a western blot of the RNA pulldown. We are concerned about the antibody and bands as the vimentin protein is barely detected in the input. Moreover, the quantification in Fig 6K does not match the large variation seen in Fig 6J. The quantification issue is likely related to the low input levels of vimentin. We also have concerns about the presence of the protein in one replicate of the antisense pulldown and whether vimentin is in fact a protein partner of SLAMR. Similarly, we also have concerns about CaMKIIalpha and detection of the protein to similar levels in one of the antisense pulldown replicates.

Reviewer #2 (Remarks to the Author):

This manuscript follows a bioinformatics-based finding on learning-related changes in expression of a lncRNA in CA1, to test the localization, trafficking, and RNA/protein interactions of the lncRNA. The authors present data that indicate that mobilization of SLAMR-containing granules accompanies activity-driven synaptic (structural) plasticity, and promotes the expression of certain proteins in this context.

The authors further show that disruption of SLAMR leads to disrupted fear memory consolidation. An understanding of these lncRNA functions would represent a significant advance in understanding the biological significance of ncRNAs, and the molecular mechanisms of memory consolidation. There are a few moderate to minor concerns that need to be addressed, but doing so will strengthen an already interesting and important study.

Concerns that can be addressed are as follows:

1) The authors do not place the finding of increased SLAMR expression after learning in context of prior work in this area. Recently, Delorme et al (PNAS 2021) identified a number of lncRNAs which were selectively modulated 6 h after contextual fear learning in the hippocampus of mice. Interestingly, in that study SLAMR was also identified as being upregulated - not after learning, but after a period of sleep disruption.

For example, on page 6 - "We, therefore, asked whether contextual fear conditioning (CFC), a form of associative learning resulting in robust long-lasting fear memories, can induce the expression of lncRNAs in specific neuronal populations and whether they function in different types of LTM." This is a point where the prior findings on changes in lncRNA in hippocampal neurons with exactly this same type of learning should be addressed.

Related to this point, it would be highly worthwhile to compare the DEG set found in this study with the DEG sets from that study. For example - on page 9 "Here, we identified 6 lncRNAs which showed significant changes in the C+S condition exclusively, indicating dynamic regulation of these lncRNAs related to hippocampal activity induced by CFC training in the dorsal-CA1 - some of these were already identified as regulated by contextual fear conditioning in whole-hippocampus by Delorme et al.

2) A serious concern is the reporting of bioinformatics statistics in the first half of the paper. It isn't clear whether these are nominal p values or adjusted p values (the more appropriate statistic to report for this type of analysis). My recommendation in the case of the former is that rationale for reporting of the nominal p value be provided as justification, and the discussion regarding these statistics be modulated accordingly. On the other hand, if these are adjusted p values, this should be much clearer in both the statistical reporting in the text and in the figures.

Related to this overarching issue, in the legend for Figure 1 "Proportional Venn diagrams derived from the DEseq analysis show

important transcriptional changes in the experimental group" I feel like "important" is a loaded word here. This result just refers to statistical significance.

3) (minor) Supplemental figure S1 - Sox 9 locus is not labeled on the map.

4) For the analysis related to spine morphological changes (responsive vs. nonresponsive) in Figure 3 - the analysis on this experiment seems slightly underpowered. How exactly was the threshold for "responsiveness" set here? Also, the time course is only 5 min? Is spine growth/formation expected to be so fast? How was this timecourse justified?

5) Related to the analysis described above - I feel like altogether this analysis would be far more convincing if the analysis was over a larger area. Is recruitment of SLAMR-containing granules always over short distances, or could it be longer-range? There doesn't seem to be a strong rationale for limiting the analysis over this spatial scale.

6) (minor) A general comment on the supplemental videos - these are generally useful, but if they could be consolidated to make it more straightforward for readers to view them, it would be beneficial. Similarly, spreading Supplemental figures over several pages is non-optimal.

Reviewer #3 (Remarks to the Author):

This is an exceptionally well-written and compelling article demonstrating novel and convincing findings with high impact. The mouse experiments include well-designed controls that make the SLAMR functional findings strongly supported. The data analysis is comprehensive, with clear descriptions of the statistical analysis that was performed (with a few noted exceptions). In particular, the well-designed RNAseq experiments to discover SLAMR and the functional experiments of Figure 8 with a huge effect size powerfully demonstrate that SLAMR in CA1 plays a central role in fear memory consolidation. Although some of the biochemical evidence is a bit weak, the major conclusions are very well supported by the current data and I do not recommend any further experimentation prior to publication. There are minor conclusions that should be weakened or better justified, and other points that merit further discussion.

The RNAseq experiments re-identify known relationships, validating their design. There is strong justification from the experiments that fear experience upregulates SLAMR, that SLAMR engages in a positive feedback loop with KIF5C to promote transport to the synapse upon stimulation, that SLAMR at the synapse regulates translation, structural plasticity and increased CaMKIIa phosphorylation, and finally that SLAMR in CA1 is necessary for fear consolidation.

Major points:

- 1) Decreased SLAMR in the synapse in response to siKif2c is less convincing as a trafficking effect given the 2-fold drop in the homogenate (Figure 3A), though the primary neuronal data supports a general role for KIF5C. The authors should temper their interpretation, or strengthen the justification.
- 2) The authors should more clearly describe how the reciprocity of KIF5C regulation fits into their model. If synaptic trafficking of SLAMR prevents its stimulation of KIF5C synthesis, is this a feedback loop to buffer SLAMR induction of KIF5C? Does SLAMR induction of KIF5C drive the reliance of dendritic arborization on SLAMR?
- 3) It is not formally shown that stimulated transport relies on KIF5C, just that basal transport relies on KIF5C. The authors should temper their interpretation, or strengthen the justification.
- 4) The discussion claims authoritatively that the increase in global translation with SLAMR overexpression is mediated through increased eIF2alpha and P70S6K. However, the actual data (Fig 5L) is correlative, not causal, and based on knockdown, not overexpression.
- 5) The proteomic method, as described, mostly pulls down non-specific interactors, since incubation is with gentle lysis buffer followed by crosslinking that will immortalize non-specific interactions during the stringent wash. Indeed, the protein IDs in Table S6E are primarily the most abundant proteins from the lysate. There are not enough replicates for confident identification of interactors. Vimentin and CaMKIIa are cherry-picked from partway down the list, without strong statistical justification. The blot in Fig 6F does not look at all convincing for CaMKIIa association with Sense, making the bar plot Fig 6G suspicious without all data points explicitly included. By contrast, the fragment data is convincing (Figure 7D), as is the evidence for CaMKIIa phosphorylation relying on SLAMR (Figure 6I). It would be best if the authors were clear that the proteomics are not a good or reliable data set, even though they were fortunate enough to discover a real and interesting interaction through the experiment. Importantly, any deficiencies in this data set do not detract from the major conclusions of the paper. These deficiencies just need to be properly described.
- 6) There is no description of instrumental proteomic methods, such as instrument, extent of fractionation, criteria for IDs, etc. Nat Comm requires that proteomic data be deposited, and they are, however reviewer credentials were not provided so that I could access the deposited data (password was required).
- 7) The SLAMR fragment deletion trafficking experiment is not that useful, because deletions could have many effects on SLAMR structure, and there is no control that I could see that demonstrates that these effects are vimentin or CaMKIIa mediated. Hence, the authors should either temper the interpretation of this experiment in the discussion and abstract or provide stronger justification.

Minor points:

- 1) It's unclear why the authors engage in extensive speculation on potential sox9 regulations when Figure S1N indicates that there is none.

- 2) The MS2:MCP method seems to mostly label regions of the cell without SLAMR (Figure S2A). Does this complicate the interpretation of this assay? Does this align with the relative intensities of Figure 2C and Figure 2D?
- 3) Figure 2L is unclear. What is the difference between docked and docking? Are the colors in the legend and the column plot being in opposite orders a mistake?
- 4) The difference between the right side of 4J and post-uncaging bars of 4K is unclear. It seems like the same experiment, and the means are the same, but the data points and error bars are very different.
- 5) The authors should justify why they used whole cell staining to evaluate translation increase by SLAMR overexpression (Fig 5G-I), but isolated synaptosomes to show loss of translation-associated proteins with SLAMR knockdown (Fig 5J-M).
- 6) The SLAMR pulldown RNAseq data looks like noise to me without compelling evidence to believe that any of it is specific or real. Given that the interpretation of this data is not necessary for any major conclusions of the manuscript, this is not an important concern, but the authors should consider temperating the conclusion that there is "a crucial role of this lncRNA in mitochondrial function"
- 7) Given the significant translational and KIF5C effects of SLAMR OE and KD, the authors should speculate on why the omics data did not provide insight into the molecular basis of these effects.

We are thankful to the reviewers for their constructive criticisms to improve our study. In this resubmission, we have carried out three additional experiments to determine whether : 1) SLAMR loss of function impacts translation, 2) SLAMR loss of function impacts CaMKII activity in spines in basal and stimulated conditions, and 3) whether SLAMR directly interacts with CaMKII holoenzyme. Further, we have carried out additional analyses as suggested by the reviewers and appropriately revised the manuscript.

Our new results are as follows: (1) Quantitative confocal imaging showed that RNAi mediated SLAMR loss of function results in decreased translation, (2) fluorescence life time imaging (FLIM) experiments with a CaMKII activity reporter revealed that RNAi mediated SLAMR loss of function results in decreased CaMKII activity in spines, and (3) in vitro CaMKII enzyme functional assay show that SLAMR RNA can significantly enhance substrate phosphorylation. Together, these experiments suggest that SLAMR lncRNA is a critical modulator of translation and CaMKII function. To reflect these new results, we have retitled the manuscript, "SLAMR, a synaptically targeted long non-coding RNA, promotes structural plasticity by modulating translation and CaMKII activity." These results are included in the manuscript and have updated our methods, results and discussion sections.

Below we describe a point-by-point response for all comments (in italics) provided:

Reviewer #1

The authors of the study entitled " Synaptically recruited lncRNA SLAMR facilitates structural plasticity in fear memory consolidation by modulating translation" use behavioral training models and compare the differential gene-expression changes before and after training. This leads to the identification of SLAMR (RikD17) that is unregulated and localized to hippocampal cytoplasm - relative to nuclear. The authors then use Fish in primary hippocampal neurons and find punctate staining at dendrites. The authors then demonstrate that the mouse lncRNA translocates in the RAT neurons. It is a bit surprising that there is no discussion of isolating the rat version of this lncRNA before using mouse in an orthotopic manner. The lncRNA locus is syntactic in RAT (via a quick blast of mouse D17) therefore would seem better suited for a homotopic analyses rather than heterotrophic. Nonetheless the authors find that there is a lot of movement of RNA granules after training than before. However, these don't seem to be specific to dendrites and rather movement in all directions. Thus the thread is a bit confusing of a dendrite localized RNA in mouse, testing its movement in rat and finding that granules are moving in all directions. The authors further test a couple of proteins by knockdown and find KIF5C seems to mediate mouse SLAMR localization in rat neurons. The authors further show that loss of SLAMR decreases KIF5C and over-expression increased KIF5C. The authors also show that loss and gain of SLAMR results in less or more arborization in neurons, respectively. The authors then pull down SLAMR, unclear if in mouse or rat neurons, and find associated RNAs and proteins. They then further parse this down to a 200nt fragment that binds CaMKIIa and Vimentin. They then determine if the 200nt mouse region affects localization in rat neurons -- removing the 200nt binding regions results in less mobility. Finally, the authors test the role of SLAMR in memory and recall etc. They used Gapmers to deplete what I think is mouse SLAMR in mouse neurons. They find SLAMR important for memory consolidation, but not other aspects tested. The authors further find that SLAMR is not relevant for other hippocampal functions.

Overall there is a lot of data in this study. However, the function of SLAMR remains unclear. There are characterizations of the mouse SLAMR in rat neurons. This is where most of the "function" is described. It is less clear that SLAMR has a clear function in mouse where I believe the original DEG were identified. There are so many quilted parts of data is very hard to

determine how things fit together and or what SLAMR does in the context of training where it was discovered. I found that the mouse SLAMR-ms2 studies were not very convincing in relationship to the training where the authors identified SLAMR. Do all RNAs move more during training? If so then SLAMR would just be one of many mobile RNAs during this process.

I am concerned that a general reader would not follow the experiments nor how to continue to study SLAMR in the context of learning -- other than generating a mouse knockout and/or GOF model. I feel this study would be best suited in a learning neuroscience focused journal -- rather than the large and general readership of nature communications.

We thank you for this feedback. We understand the concerns indicated by the reviewer regarding some critical points of this study. We have tried to improve the previous version by carrying out three new sets of experiments to determine whether : 1) SLAMR loss of function impact translation, 2) SLAMR loss of function impact CaMKII activity in spines, and 3) whether SLAMR directly interact with CaMKII holoenzyme. We also added additional analyzes and have carefully revised the manuscript.

As this reviewer noted, SLAMR lncRNA was discovered as a part of an unbiased screen of dorsal CA1 enriched RNAs in response to contextual fear conditioning. An exciting point regarding its upregulation is that SLAMR enrichment is specific to CA1 when compared to CA3 neurons. While characterizing its function, the finding that it is localized to dendrites prompted us to closely examine its localization mechanisms. We resorted to rat neurons because of the technical advantages it offered for MS2 based tracking. The relation with KIF5C, and the subsequent characterization were done in mouse neurons. However, to obtain detailed transport parameters, we utilized rat neurons. All RNAs are not targeted to dendrites, and lncRNA targeting to dendrites is poorly understood. Most importantly, lncRNAs entering into spine compartment has not been shown before. Our quantitative imaging documented in this study revealed that SLAMR is actively transported in dendrites, enters spine compartments, and move towards stimulated spines. All of these are exciting and novel finding for lncRNA biology.

Functionally, we find that SLAMR modulates translation. Gain of function increases translation whereas its loss of function decreases translation (new data-Figures 5J,K and Supplementary Figure S4B). Importantly, there is a corresponding change in arborization and spine morphology with loss and gain of function of SLAMR. The detailed characterization of SLAMR interactome, not only resulted in identification of a critical element in SLAMR that mediates its interaction but also identified CamKII alpha as a protein that interacts with. We set up fluorescence lifetime imaging assays to examine the dependence of CaMKII function on SLAMR expression levels using mouse organotypic slices with neurons expressing SLAMR RNAi and a reporter for CaMKII activity (Anant, J., et al. (2023), bioRxiv) . This imaging has shown that SLAMR RNAi results in lower basal CaMKII activity in dendrites and an overall reduction in activity even with stimulation (Figure 7 A,C,E). We next tested if SLAMR can directly impact CaMKII activity in vitro by quantifying substrate (syntide) phosphorylation by CaMKII in the presence and absence of SLAMR. Our results suggest that CamKII activity is enhanced by the presence of SLAMR Sense lncRNA in the kinase reaction mix (New data-Figure 7G,H), suggesting a direct interaction with CamKII.

We request this reviewer to kindly consider our novel findings on SLAMR regulation in CA1, its impact on translation, CaMKII activity, structural plasticity and our findings on its active localization and underlying mechanisms, unbiased determination of its interactome providing novel biology of lncRNAs and its role in specific memory processes.

Furthermore, please consider that the machinery supporting the transport of SLAMR is present in both rat and mouse models. Especially considering that this lncRNA has orthologs in

previously described species like LINC00673 in humans and LOC110366352 (SlincR) in zebrafish. Interestingly, rat locus also seems very well conserved with similar positions for an important number of genes including Sox9, Cog1, Sstr2, and Slc39a11. In this sense, the lncRNA LOC102549836, a lncRNA with unknown functions, shows an equivalent position to SLAMR with the potential to bind Sox9 promoter. LOC102549836 also displays higher expression in the brain (BioProject PRJNA238328) compared to all other organs making it a potential ortholog of SLAMR in rat. Additionally, most lncRNAs show poor conservation in their sequences between species, nevertheless, their functions can be conserved based on their secondary structures, something common for lncRNAs. We included a discussion of these points in the manuscript to clarify this important aspect of the study and indicated more clearly the experiments carried out on mice or rats in the main text. Previous works demonstrated that several mRNAs from critical genes involved in learning and memory processes increase their expression/transport in distal parts of the neurons, like the calcium calmodulin members CaMK2N1 or CaMKII α (Ex. Ortiz et al., 2017, PMID: 28683307; Astudillo et al., 2020, PMID: 32320502), as an example. Only those RNAs involved or required for these functions show movement along the dendrites. However, this phenomenon does not mean that all particles displaying directed transport in dendrites necessarily have more relevance for early plasticity than others. Importantly, our results demonstrate a specific increase in SLAMR lncRNA recruitment into the stimulated spine, which is completely novel and has not been reported in any system before.

Here are our responses to specific comments.

1) Fold changes and Pvalues should be listed for the reader to understand the effect sizes being measured. For example: RNAseq data showed a significant reduction of Sox9 in the dorsal CA1 after CFC training exclusively in the C+S condition compared to both control groups. How much and how significant were the changes - should be listed in main text for any claim of "significance"

We would like to thank the reviewer for their observations regarding the statistical analysis. We included most of this information in the supplementary tables (RNAseq is located in Supplementary table 1). However, we understand that the most important pieces of this information should also be included in the main text. We reviewed the manuscript very carefully to improve this aspect and include all the missing information in the results section. For the example mentioned, we indicated in the results section that "Sox9 is significantly reduced in dorsal CA1 after CFC in the C+S condition compared to Context alone (log₂FC=-0.679; pVal=0.0055) and shock alone (log₂FC=-0.677; pVal=0.0054)".

2) There are no statistics or quantification of "dendritic" localization in Figure 1J. This should be quantified and determined if significant before training as well.

Thank you for this suggestion. Images of FISH included in Figure 1 are primary hippocampal mouse neuronal cultures without any kind of stimulation. FISH conditions represented in this figure include D17Rik-probe compared with the 'no probe' condition to test for non-specific labeling. The 'no probe' condition shows that there is no unspecific binding from the fluorescent label. Primary hippocampal neurons are collected from E15 to P1 pups for dissection and plating and maintained for two weeks for full development. As hippocampal neurons lose the ability to be cultured as mice develop, it is not possible to carry out this experiment in mice after

or just before training. The main objective of this experiment was to verify the subcellular localization of SLAMR lncRNA and confirm the fractionation results.

3) *"We used an shRNA plasmid to
471 constitutively silence the expression of SLAMR and alternatively, another one under a
472 doxycycline-inducible promoter (TET) for temporal control of SLAMR KD. A plasmid
473 expressing a scrambled sequence was used as negative control (NC). All plasmids also
474 express eGFP to visualize transfected neurons and assess morphological changes
475 (Figure 4A,B)."*

It is very confusing in many places in the manuscript if the neurons are RAT-based or Mouse, considering mouse SLAMR is being tested in rat. For example, is the shRNA to an ectopic ally expressed mouse SLAMR in rat neurons? Also, what is the effect size of SLAMR on "Arborization" of neurons in the experiments examined in text pasted above? For example, Figure 4 H-K shows only very small effects (10% to 15% in K etc...

Thank you for this comment. We regret the lack of clarity. We have now reviewed the manuscript carefully and clarified in the main text and figure legends whether the experiments were developed in rat (lncRNA transport) or mouse (all other analyses) neurons.

Rat neurons were transfected only with MCP plasmids, a coat protein that recognizes the hairpin loops and is functional in many cell types and species; and MS2-SLAMR plasmids, which introduces the SLAMR RNA, tagged with 32-hairpin loops, in the rat cell and use its machinery for transport.

The arborization analysis (Figures 4B,C and Figures 5B,C) was conducted in mouse neurons and measured by Sholl analysis. Sholl analysis is one of the most common methods to analyze morphological changes in neurons related to dendritic development ("arborization"). The arborization (number of intersections) of neurons transfected with SLAMR-KD and TET-SLAMR is significantly lower compared to neurons transfected with the NC-GFP control plasmid starting at 30 μm from the soma until 100 μm , where we established our limit. The values of significance show variations with the distance, something common in this type of study, and a global analysis of effect size using the coefficient of d_{Cohen} resp. g_{Hedges} indicates a large effect for both experimental conditions compared to NC-GFP control condition (SLAMR-KD= 2.243; TET-SLAMR=2.394) (See statistical information in Supplementary Table S4)

The experiments represented in Figure 4 H-K are related to glutamate uncaging assays and evaluate the effect of SLAMR silencing on the volume of the spines after the induction of structural long-term potentiation (sLTP), but they do not measure changes in dendritic arborization. The results of these experiments help us to understand how the loss of SLAMR in the cells alters spine functionality (in terms of structural plasticity) rather than describe morphological changes. Also, Fast rate (Fig. 4M) and slow rate transient (Fig. 4J) imaging show significant differences between the NC-GFP (~5-15% volume change) condition and TET-SLAMR (~50-60% volume change) condition. This suggests that SLAMR is critical in facilitating the rapid structural changes in response to stimulation, required for memory formation. However, the degree of sustained growth represented in Fig. 4K is negligible between NC-GFP and SLAMR as indicated by the reviewer; these results do not show any significant differences between groups (Fig. 4J(sustained)), only between pre- and post-stimulation for each condition (Fig 4K). This suggests that SLAMR functions in the initial phases of structural plasticity but not in maintaining spine volume.

4) The "movement" and arborization studies are very confusing in RAT neurons. I am not sure how any of these findings relate back to how the RNA was discovered. I would suggest testing behavioral studies in a knockout model. Are the learning differences noticeable? The knockdown studies are not very convincing, and most are found to be negative. It is also hard to understand how longer-term responses could be affected by acute timeframes of knockdown (48-72 hrs.).

As we mentioned before, only transport-related experiments were carried out using rat neurons, while characterization, morphology, function, and mechanistic studies were performed using mice primary cultures and mice models.

In our study, we discovered a novel lncRNA that is enriched in CA1 neurons after CFC, whose function was never described before. Previous studies indicate that subcellular localization (particularly nuclear vs. cytoplasmic) is crucial in determining the mechanism of action of lncRNAs. Interestingly, our FISH results in mouse primary hippocampal neurons demonstrate that SLAMR was localized in the cytoplasm as well as in the dendrites (Fig. 1J). Dendritic localization of lncRNAs is not common and probably indicates that this lncRNA is needed to function in distal parts of the neuron. This is why it was crucial for our study to include and analyze the mechanism of transport of SLAMR. In this regard, transport and arborization studies are intimately related. Arborization is a complex process critical for learning and memory functions that implicates different mechanisms of transport (ex. kinesins) and involves a high variety of cargoes (ex. proteins, mRNAs), among others. For this reason, we consider that transport and morphological studies are decisive in explaining why the enrichment of this lncRNA is required in the early stages of memory acquisition.

Indeed, mice knockout models can be useful in studying cognitive functions using behavioral tests. However, currently, we do not have sufficient funding for generating a knockout model, and generating this model would require many months. Also, our results indicate that constitutively SLAMR silencing using shRNAs induces a dramatic decrease in dendritic arborization, so we cannot discard the possibility that the absence of this lncRNA may affect the early stages of development in mice.

Lastly, we consider that the lack of an effect in many behavioral tasks suggests the specificity and strength of our study. It is well known that behavior training involves rapid changes in transcriptome and translation that ultimately result in structural changes required for long-term memory storage. The impairments we see in memory consolidation but not acquisition in the knockdown condition suggest that SLAMR may participate in these early transcriptional and translational changes. We do not know the requirements of these molecular changes in extinction, recall, and spatial memory. To demonstrate the rigor of our finding, we also assessed the impact of SLAMR loss of function in CA3. We request this reviewer to kindly consider all our efforts to carry out in-depth experiments that ultimately suggest specificity. From the 6 independent *in vivo* manipulations, we concluded that SLAMR specifically impairs the consolidation of contextual fear.

Other major considerations:

A) Figure 1, Lines 173-180, 1017-1036, 1526-1544. The authors fail to mention how many animals were sacrificed for each condition i.e., sample size (n=) for the RNAseq.

We really appreciate this observation. The missing information for RNAseq (CA1 CFC) and qRT-PCR (CFC CA1 & CA3, and MWM CA1) was included in the sections indicated by the reviewer.

B) MS2-MCP experiments in Figure 2, 3B-G, 7D, and on. The authors studied the activity and protein binding partners of murine lncRNA in the context of rat neuronal hippocampal cultures, in other words, they studied the function and RNP structure of a lncRNA in a different species without initial or similar experiments in the original species. While some lncRNA may act similarly in different species, the seminal experiments on a lncRNA's function should be mainly performed in the species of its origin, in this case, mouse. The authors found possible homologues in human and zebrafish but found none in rat. We question the validity and conclusions of these experiments considering this and whether function in a different species is significant and relevant, without first studying it in the context of its original species, mouse. The rat cell experiments must be repeated in mouse cells. We also stress the importance of indicating the cell species in their figures and figure legends.

As we mentioned before, the reason for the use of rat hippocampal cultures for imaging in transport studies is related to technical requirements. Importantly, the machinery supporting the transport of SLAMR is present in both rat and mouse models. Especially considering that this lncRNA has orthologs in an important number of other species, including humans, as well as possible conserved functions based on its secondary structure rather than the specific sequence, something common for lncRNAs. Regarding the conservation of this lncRNA in rats, we initially focused our discussion on those orthologs (human and zebrafish) that were already described and discussed by previous studies. The lncRNA LOC102549836, a lncRNA with unknown functions, shows an equivalent position to SLAMR with the potential to bind Sox9 promoter as well as an excellent expression in the brain (BioProject PRJNA238328) becoming a potential ortholog of our lncRNA of interest. However, we consider that analyzing the functional conservation of all these orthologs is not the main objective of this study and will be expensive in terms of time and effort. Besides, we agree with the reviewer that this important point needs some clarification, and we follow the reviewer's recommendations to improve the identification of the species used for each experiment.

In addition, we would like to highlight that all the main experiments related to SLAMR protein interactions were carried out in mice (Fig. 6, Fig. 8A-F). Experiments included in Figure 2 analyze only the transport of SLAMR along the dendrites. Figure 3, suggests the possible role of KIF5C in SLAMR transport, analyzed both in mouse neurons (Fig. 3A, H-I) and in rat neurons (Fig. 3C-G). Our results show that there is a reciprocal regulation of these two elements in mouse neurons, which could take place through an indirect mechanism instead of a direct binding. Also, Figure 8D refers to the study of CaMKII α and Vimentin interactions with SLAMR using a pull-down strategy, for this experiment we use lysates from mouse hippocampal tissue. Furthermore, Figure 8G-L analyzes if the fragment that interacts with CaMKII α and Vimentin is also involved in the transport, in this case, using rat neuronal cultures but focusing on transport conditions rather than protein interactions.

C) The authors use two different species of primary hippocampal cells in their manuscript, however there are several examples of noting the cells, but not the species. The authors must clearly indicate the species of primary hippocampal cells throughout the manuscript (e.g., Lines 378-379, Figure 3A, 420-421, 442, Figure 3H-K, 444, 524, 545, 554, Figure 4A, Figure 5A, etc.).

Thank you for pointing this out. In this revised version of the manuscript, we specify all the details regarding the type of sample used for each experiment, especially in the results section indicated by the reviewer.

D) Figures 6F, 6J, and 6K. The authors identified protein partners of SLAMR via mass spec and sought to validate using a western blot of the RNA pulldown. We are concerned about the antibody and bands as the vimentin protein is barely detected in the input. Moreover, the quantification in Fig 6K does not match the large variation seen in Fig 6J. The quantification issue is likely related to the low input levels of vimentin. We also have concerns about the presence of the protein in one replicate of the antisense pulldown and whether vimentin is in fact a protein partner of SLAMR. Similarly, we also have concerns about CaMKIIalpha and detection of the protein to similar levels in one of the antisense pulldown replicates.

Western-blot validation of pull-down experiments sometimes shows this kind of pattern, especially for scarce proteins like vimentin (see our previously published study as an example: Grinman et al., 2021). Vimentin is enriched in the brain during development however its levels are low in adults. Nevertheless, our results show that the binding to SLAMR is strong enough to be pulled down with SLAMR probes after incubation with fresh hippocampal mouse lysates. Proteins show more enrichment in the sense of the pull-down condition than input and occasional faint bands can appear in the antisense condition. The purification product is not perfect. For this reason, these experiments were repeated at least 4 times in different extracts of different mice with 3 technical replicates for each condition. The quantification plot observed is the average of all the biological replicates and the western-blot picture included is only a representative image. Also, the pulldown with SLAMR probe in Figure 8 A-F, using the protected fragment of SLAMR that seems to interact with both proteins, shows stronger enrichment, clearer results, and supports our previous findings from Figure 6.

In summary, we sincerely hope that this reviewer is convinced about the novelty of our work, satisfied with our responses to critiques and new data we included to explain SLAMR function, and support the publication of our study.

Reviewer 2#

This manuscript follows a bioinformatics-based finding on learning-related changes in expression of a lncRNA in CA1, to test the localization, trafficking, and RNA/protein interactions of the lncRNA. The authors present data that indicate that mobilization of SLAMR-containing granules accompanies activity-driven synaptic (structural) plasticity, and promotes the expression of certain proteins in this context. The authors further show that disruption of SLAMR leads to disrupted fear memory consolidation. An understanding of these lncRNA functions would represent a significant advance in understanding the biological significance of ncRNAs, and the molecular mechanisms of memory consolidation. There are a few moderate to minor concerns that need to be addressed, but doing so will strengthen an already interesting and important study.

Concerns that can be addressed are as follows:

- 1) *The authors do not place the finding of increased SLAMR expression after learning in context of prior work in this area. Recently, Delorme et al (PNAS 2021) identified a number of lncRNAs which were selectively modulated 6 h after contextual fear learning in the hippocampus of mice. Interestingly, in that study SLAMR was also identified as being upregulated - not after learning, but after a period of sleep disruption. For example, on page 6 - "We, therefore, asked whether contextual fear conditioning (CFC), a form of associative learning resulting in robust long-lasting fear memories, can induce the expression of lncRNAs in specific neuronal populations and whether they function in different types of LTM." This is a point where the prior findings on changes in lncRNA in hippocampal neurons with exactly this same type of learning should be addressed. Related to this point, it would be highly worthwhile to compare the DEG set found in this study with the DEG sets from that study. For example - on page 9 "Here, we identified 6 lncRNAs which showed significant changes in the C+S condition exclusively, indicating dynamic regulation of these lncRNAs related to hippocampal activity induced by CFC training in the dorsal-CA1 - some of these were already identified as regulated by contextual fear conditioning in whole-hippocampus by Delorme et al.*

Thank you for this comment. We agree with the reviewer that despite the lack of knowledge about this lncRNA, some previous studies that published hippocampal transcriptomic profiles must be explored and included in the study. However, we do not want to forget to mention that research studies with similar objectives and methodology are scarce and most of the previously published studies did not show parallelism with our methodology. In any case, we reviewed RNAseq results from similar studies using hippocampal tissue to have a bigger picture of the complex regulation of SLAMR and improve the discussion. This information was included in our discussion considering the Delorme et al. study. As the extension of the introduction is limited and we do not mention SLAMR until the end of this section to summarize our study, we placed these references in the discussion section.

2) A serious concern is the reporting of bioinformatics statistics in the first half of the paper. It isn't clear whether these are nominal p values or adjusted p values (the more appropriate statistic to report for this type of analysis). My recommendation in the case of the former is that rationale for reporting of the nominal p value be provided as justification, and the discussion regarding these statistics be modulated accordingly. On the other hand, if these are adjusted p values, this should be much clearer in both the statistical reporting in the text and in the figures.

Related to this overarching issue, in the legend for Figure 1 "Proportional Venn diagrams derived from the DEseq analysis show important transcriptional changes in the experimental group" I feel like "important" is a loaded word here. This result just refers to statistical significance.

We appreciate this comment and agree with the observation that the description of the transcriptomic analysis must be revised. We have now improved the description of the methodology followed in our analysis by adding all the missing information, especially the adequate rationale for reporting nominal p values as well as changing ambiguous expressions like "important changes" referring to statistical significances.

3) (minor) *Supplemental figure S1 - Sox 9 locus is not labeled on the map.*

Thanks for pointing this out. Sox9 labeling was missing in this figure. .

Upon revision of the manuscript and in order to fit related supplemental materials into one figure, as suggested by this reviewer in minor point 6, we removed this figure. We did not think it provided additional clarity to the general audience of Nature Comm. Additionally, the locus that was depicted in the map can easily be found on NCBI by a reader with more interest in understanding the genomic localizations of lncRNA.

4) *For the analysis related to spine morphological changes (responsive vs. nonresponsive) in Figure 3 - the analysis on this experiment seems slightly underpowered. How exactly was the threshold for "responsiveness" set here? Also, the time course is only 5 min? Is spine growth/formation expected to be so fast? How was this time course justified?*

We understand the concerns described in this question. The methodology followed in these experiments was based on previous publications with well-established protocols, including studies from our collaborators as well as from our team (Matsuzaki et al., 2004; Saneyoshi et al., 2019, Grinman et al., 2021). Based on previously established criteria, we consider as "Responsive Spines" those that increase by at least 10% of their volume and do not remain stagnant or shrinking. This is a process that usually takes place very fast in neurons. For example, results in Figure 4H-I show how the volume of the spines registered has a peak at 1.5 min and begins an initial plateau at 5 min, we observed something similar in our previous studies (Grinman et al., 2021). We improved the results section and addressed all these important questions in the current version of the manuscript.

5) *Related to the analysis described above - I feel like altogether this analysis would be far more convincing if the analysis was over a larger area. Is recruitment of SLAMF-containing granules always over short distances, or could it be longer-range? There doesn't seem to be a strong rationale for limiting the analysis over this spatial scale.*

Thank you for this comment. We would first like to note that even though dendrites are relatively thin they have many different focal planes, and we were limited to the focal plane that also included the stimulated spine. We are not able to take z-stacks during this type of experiment due to the temporal resolution needed to capture transport. With these limitations, we found that ~25µm on either side of the stimulated spine was the maximal distance that would reliably be in the same focal plane. The protocol selected for this experiment was also previously optimized and published by prestigious laboratories with a long experience in this field (Bauer et al., 2017, 2019). Furthermore, these experiments were meant to capture whether or not SLAMF could be recruited to stimulated spines, a lncRNA behavior never described before. We do think it is possible and indeed probable that in the mouse, over time, we would see more movement from distal regions (certainly from the nucleus) after contextual fear conditioning, where clusters of spines rather than single spines would undergo structural plasticity. We are interested in exploring these possibilities in more detail in future studies.

6) (minor) A general comment on the supplemental videos - these are generally useful, but if they could be consolidated to make it more straightforward for readers to view them, it would be beneficial. Similarly, spreading Supplemental figures over several pages is non-optimal.

Thanks for this comment. We have combined movie files from the same analysis into one movie, reducing the number of movie files from 24 to just 9. We have also added labels to the dendrites in the movies so the reader can easily make comparisons between different conditions. Also, supplemental figures that were separated into several pages are now unified in a single figure for optimization.

We hope that this reviewer finds that the addition of new data, revisions and our responses to the critiques are satisfactory and support publication of our study.

To Reviewer #3:

This is an exceptionally well-written and compelling article demonstrating novel and convincing findings with high impact. The mouse experiments include well-designed controls that make the SLAMR functional findings strongly supported. The data analysis is comprehensive, with clear descriptions of the statistical analysis that was performed (with a few noted exceptions). In particular, the well-designed RNAseq experiments to discover SLAMR and the functional experiments of Figure 8 with a huge effect size powerfully demonstrate that SLAMR in CA1 plays a central role in fear memory consolidation. Although some of the biochemical evidence is a bit weak, the major conclusions are very well supported by the current data, and I do not recommend any further experimentation prior to publication. There are minor conclusions that should be weakened or better justified, and other points that merit further discussion.

The RNAseq experiments re-identify known relationships, validating their design. There is strong justification from the experiments that fear experience upregulates SLAMR, that SLAMR engages in a positive feedback loop with KIF5C to promote transport to the synapse upon stimulation, that SLAMR at the synapse regulates translation, structural plasticity and increased CaMKIIa phosphorylation, and finally that SLAMR in CA1 is necessary for fear consolidation.

Major points:

1) *Decreased SLAMR in the synapse in response to siKif2c is less convincing as a trafficking effect given the 2-fold drop in the homogenate (Figure 3A), though the primary neuronal data supports a general role for KIF5C. The authors should temper their interpretation, or strengthen the justification.*

Thank you for pointing this out, we included the respective clarifications in both sections, results and discussion. As noted, Figure 3A describes the effect of silencing some of the main kinesins in the global expression and synaptic presence of SLAMR. Silencing **KIF2A** significantly increases (*p<0.05) the presence of SLAMR in the isolated synaptoneuroosomes but it does not change its expression in the total homogenate. On the other side, **KIF5C** silencing induces a significant decrease of SLAMR in total homogenate (*p<0.05) and especially in the synaptoneurosome-isolated fraction (**p<0.01). However, we understand that considering the basal levels of SLAMR (siNC) in the synaptic fraction, the decrease in the expression of this

lncRNA in the synaptoneurosome after siKif5c (despite its significance) may not be enough convincing. So, we decided to temper the explanation regarding the fractionation study and focus our discussion on transport experiments to explain the role of KIF5C in SLAMR regulation.

2) The authors should more clearly describe how the reciprocity of KIF5C regulation fits into their model. If synaptic trafficking of SLAMR prevents its stimulation of KIF5C synthesis, is this a feedback loop to buffer SLAMR induction of KIF5C? Does SLAMR induction of KIF5C drive the reliance of dendritic arborization on SLAMR?

We agree with the reviewer that this relationship needs more clarification and we have now incorporated this into the discussion. Our results show that silencing Kif5c synthesis reduces the general expression of SLAMR (Fig. 3A) as well as its synaptic trafficking (Fig. 3B-G). Similarly, SLAMR silencing decreases global Kif5c expression while SLAMR overexpression increases Kif5c levels (Fig. 3H-I). These results indicate a positive correlation between these two elements with a reciprocal correlation. We do not have evidence of the opposite effect or anything that suggests that synaptic trafficking of SLAMR prevents the stimulation of KIF5C synthesis.

Also, we found some parallelism in dendritic arborization when SLAMR or KIF5C are overexpressed or silenced. As we observe in our study (Fig. 4 A-C) and previous works from our lab (Swarnkar et al., 2021) silencing the expression of either KIF5C or SLAMR induces a significantly decrease in dendritic arborization. Further, overexpression of SLAMR (Fig. 5A-C) or KIF (Swarnkar et al., 2021) has the opposite effect, increasing the dendritic arborization. We see similar parallelism in regard to both SLAMR (Fig. 5G-O) and KIF5C's (Swarnkar et al. 2021) roles in enhancing translation.

Taken together all these results support the idea of a reciprocal regulation between these two elements. Our results indicate that regulation of dendritic arborization by SLAMR requires the motor protein KIF5C for its function, and SLAMR is part of KIF5C's cargoes required for dendritic arborization. An increase in SLAMR levels in the cell will also increase the KIF5C expression required for this transport. Conversely, a decrease in the availability of KIF5C will be followed by a decrease in SLAMR levels consequently.

3) It is not formally shown that stimulated transport relies on KIF5C, just that basal transport relies on KIF5C. The authors should temper their interpretation, or strengthen the justification.

Thank you. Due to technical limitations, stimulated transport represented in Fig. 3K-H was exclusively focused on SLAMR transport. These experiments already require specific constructs (SLAMR-MS2) and double transfections (MCP) that implicate complex execution. All the transport experiments included in this study were performed under the same protocol that was previously optimized under basal conditions, also following the recommendations of previous studies (Bauer et al., 2017, 2019). Nevertheless, we agree with the comment and tempered our description of KIF5C's role in the transport of SLAMR in stimulated neurons in the discussion.

4) The discussion claims authoritatively that the increase in global translation with SLAMR overexpression is mediated through increased eIF2alpha and P70S6K. However, the actual data (Fig 5L) is correlative, not causal, and based on knockdown, not overexpression.

Thanks for pointing this out. First, we found that, as the reviewer mentioned, this experiment was incorrectly described in the discussion. We have fixed this mistake in this newer version.

Also, we temper our claim regarding the role of SLAMR acting through eIF2 α and P70S6K in global translation.

5) The proteomic method, as described, mostly pulls down non-specific interactors, since incubation is with gentle lysis buffer followed by crosslinking that will immortalize non-specific interactions during the stringent wash. Indeed, the protein IDs in Table S6E are primarily the most abundant proteins from the lysate. There are not enough replicates for confident identification of interactors. Vimentin and CaMKII α are cherry-picked from partway down the list, without strong statistical justification. The blot in Fig 6F does not look at all convincing for CaMKII α association with Sense, making the bar plot Fig 6G suspicious without all data points explicitly included. By contrast, the fragment data is convincing (Figure 7D), as is the evidence for CaMKII α phosphorylation relying on SLAMR (Figure 6I). It would be best if the authors were clear that the proteomics are not a good or reliable data set, even though they were fortunate enough to discover a real and interesting interaction through the experiment. Importantly, any deficiencies in this data set do not detract from the major conclusions of the paper. These deficiencies just need to be properly described.

We do understand the concerns exposed in this question regarding the proteomics results. The protocol followed for this experiment was selected after a very long period of optimization and also used and published in previous studies from our lab (Grinman et al., 2021). In fact, the list of proteins shown in Table S6E indicates a larger number of unique proteins in the sense condition completely absent in the antisense control condition. Also, most of these specific proteins in sense condition have a weight around 35-64KDa matching with the size of the bands observed in sense replicates of the representative gel stained with silver kit in Supplementary Figure 5C. Several proteins from this list were tested by western blot to verify the interactions suggested by LC-MS/MS results. However, we selected those that showed more consistent results and the biggest expression comparing the Sense condition vs. the Antisense condition between different technical and biological replicates. The quantifications represented in Figure 6 are composed of 4 independent pull-downs with 3 samples each. We want to thank the reviewer for the observation on plot 6G, as the bar was representing SD instead of SEM by mistake (similarly in Figure 6G). We have fixed this mistake and changed the style of the plots including independent dots. And we have done the same for Figures 7 E & F (now figures 8 E,F). In any case, following reviewer's recommendations we tempered our arguments regarding the LC-MS/MS results and focused on protected fragments and CaMKII α phosphorylation.

6) There is no description of instrumental proteomic methods, such as instrument, extent of fractionation, criteria for IDs, etc. Nat Comm requires that proteomic data be deposited, and they are, however reviewer credentials were not provided so that I could access the deposited data (password was required).

We agree with the reviewer that the methods section related to proteomic studies were a little scarce. The current version of the manuscript includes a more detailed and extended version of the methodology followed.

Additionally, proteomic data was deposited to MassIVE with the project number: MSV000091477. We apologize if the instructions for access were not clear. You can use the Username: MSV000091477_reviewer & Password: "a" to access this information.

7) The SLAMR fragment deletion trafficking experiment is not that useful, because deletions could have many effects on SLAMR structure, and there is no control that I could see that demonstrates that these effects are vimentin or CaMKII α mediated. Hence, the authors should either temper the interpretation of this experiment in the discussion and abstract or provide stronger justification.

Thank you for this suggestion. Indeed, it is well-known that different variations in lncRNAs structure can induce dramatic changes in their function and interactions. The objective of this experiment was to explore the critical role of the fragment removed in determining the function and transport of SLAMR. We think that CaMKII α or Vimentin do not play any direct role in SLAMR transport, besides, the results indicated that the fragment removed is important for those protein interactions as well as for the transport. We seriously considered the reviewer's comment regarding this point and tempered our arguments.

Minor points:

- 1) It's unclear why the authors engage in extensive speculation on potential sox9 regulations when Figure S1N indicates that there is none.*

We made this decision based on previous studies from humans and zebrafish (Li et al., 2017; Garcia et al., 2017) extended by their authors on their mouse ortholog. These studies suggested that SLAMR has the potential to regulate Sox9 expression due to the proximity between their promoters. As this is the only speculated function described for SLAMR, we thought that it was an important aspect to explore. However, as the reviewer indicates, our results demonstrate that SLAMR manipulations did not induce any change in the expression of Sox9. In any case, based on the previous hypothesis suggested regarding the function of this lincRNA, we decided to maintain these data. While it is probably not critical for our study, it is important as a rebuttal to these previous studies.

- 2) The MS2:MCP method seems to mostly label regions of the cell without SLAMR (Figure S2A). Does this complicate the interpretation of this assay? Does this align with the relative intensities of Figure 2C and Figure 2D?*

Our results indicate the presence of some MCP-GFP positive particles outside the nucleus in both soma and dendrites (Sup Fig 2A). However, as Figure 2C shows, these particles show no movement along the dendrites. On the other side, most of the particles detected in the MS2-SLAMR:MCP condition showed movement in a 5 min time frame as the kymographs indicate. Kymographs represented in Figures 2C and 2D do not represent the presence of particles in terms of "intensity", they show the presence of movement and their direction.

In addition, there are a couple of factors that can influence this kind of result. First, the probe does not distinguish between MS2-SLAMR and endogenous SLAMR, and second, the images were taken in different types of microscopes with different techniques (Live imaging for transport vs. Fixed cells for FISH).

- 3) *Figure 2L is unclear. What is the difference between docked and docking? Are the colors in the legend and the column plot being in opposite orders a mistake?*

Thanks for this observation. We clarify this point in the new version of the manuscript. We define as “Docked” those particles that remained stationary at a spine during the entirety of the movie and “Docking” means that during the recording we witness SLAMR becoming stationary at a spine. The colors indicated in the legend were in a different order than the plot. The mistake was fixed for the current version of the manuscript.

- 4) *The difference between the right side of 4J and post-uncaging bars of 4K is unclear. It seems like the same experiment, and the means are the same, but the data points and error bars are very different.*

The original data source is the same slow-rate imaging for Fig. 4J and Fig. 4K, but different analyses were carried out to evaluate different aspects of the experiment. Specifically, Fig. 4J represents single time point comparisons between negative control (NC, n=12) and induced silencing by TET of SLAMR (TET-SLAMR, n=13) at +1.5 min or +30.5 min. Data points included in this plot are which are in each transient and sustained phase that represent only single time points show as "+1.5 min" or "+30.5 min" showing a statistical difference between NC and TET-SLAMR.

On the other side, Fig. 4K represents Pre- vs Post-uncaging comparisons within NC and TET-SLAMR groups based on five-time points. This figure has five data points in each group which represent five different time points before (Pre, -4.5, -3.5, -2.5, -1.5, -0.5 min) and after (Post, +28.5, +29.5, +30.5, +31.5, +32.5 min) the first uncaging pulse acquired with the slow-rate imaging protocol. Then, the ranges of five-time points are shown as bars ("Pre" and "Post") in Fig. 4I. The results of this analysis represent whether uncaging induces prolonged volume increases in both the NC and TET-SLAMR groups.

- 5) *The authors should justify why they used whole cell staining to evaluate translation increase by SLAMR overexpression (Fig 5G-I), but isolated synaptosomes to show loss of translation-associated proteins with SLAMR knockdown (Fig 5J-M).*

These are two different approaches required to evaluate both global changes and specific functions in spines. As the reviewer indicates, Puromycin experiments are very useful to evaluate global changes in translation, as well as observe these specific changes in the cell body or dendrites thanks to the labeling. However, it is not a procedure recommended to observe specific changes in small structures like synapses for technical issues. The isolation of synaptoneurosome is a specific technique highly recommended to evaluate specific changes in translational machinery and its results are critical to understanding the reason why our lncRNA is required to be transported to these structures. These experiments are complementary and are necessary to elucidate the specific role of SLAMR in translation.

Additionally, we have since added a more directly parallel experiment to the puromycin staining in SLAMR-OE neurons by evaluating puromycin labeling in SLAMR-KD neurons as well (Figure 5J,K and Supplementary figure S4B). We found that

SLAMR-KD significantly reduces puromycin staining in dendrites but not in the cell body. These findings strengthen the role of SLAMR in enhancing translation.

- 6) *The SLAMR pulldown RNAseq data looks like noise to me without compelling evidence to believe that any of it is specific or real. Given that the interpretation of this data is not necessary for any major conclusions of the manuscript, this is not an important concern, but the authors should consider temperating the conclusion that there is "a crucial role of this lncRNA in mitochondrial function"*

Thank you for this suggestion. We have used all the appropriate controls in the pull down experiment. Due to funding issues, we did not characterize RNA components of the complex. We found that CaMK II alpha protein as an interacting component. This interaction of SLAMR to CaMK II was validated by pull down assays in multiple independent assays using full length SLAMR as well as using fragments of SLAMR predicted to bind its interacting proteins, and by enzyme functional assays. Based on reviewers suggestion, we have tempered down the statement regarding SLAMR and mitochondrial function.

- 7) *Given the significant translational and KIF5C effects of SLAMR OE and KD, the authors should speculate on why the omics data did not provide insight into the molecular basis of these effects.*

Thank you for this suggestion. The RNA components of the SLAMR interactome could function as substrates of translation. Lack of identification of KIF5C in the complex suggests that we would need additional optimization of pulldowns such as using different detergents in pulldowns for stabilizing specific complexes. Importantly, our quantitative analysis of SLAMR transport in KIF5C RNAi neurons suggest that SLAMR transport is dependent on KIF5C function.

We hope that this reviewer is now satisfied with our responses and the revisions we made in our study (revising the manuscript, additional analyses, and three new experiments) and supports publication of our study.

REVIEWERS' COMMENTS

Reviewer #1 (Remarks to the Author):

The authors have gone to great lengths to address the questions and concerns raised as well as modify the main text. All of these responses make sense and help provide context to the results the authors are focusing on. I don't have any further questions.

Reviewer #3 (Remarks to the Author):

The authors have satisfactorily addressed all of my concerns from the prior round of review.

Reviewer #4 (Remarks to the Author):

This revised manuscript reports exciting data to support a role for a novel lncRNA, SLAMR, in dendritogenesis, spine morphology, and translation. SLAMR associates with CaMKII α protein, demonstrating a mechanism for its action in neurons. The experiments range from primary neuronal culture to neurobehavioral studies in mice. The work is of interest to a broad audience.

The authors have added a large amount of new data in response to previous criticisms, which I detail below.

Reviewer 1

In summary, they have added data to support the following:

1. SLAMR modulates translation and dendritic morphology as evidenced by both overexpression and knockdown experiments.
2. Identification of CamKII alpha as a SLAMR protein interactor.
3. SLAMR is actively transported in dendrites and spines and that this transport is regulated by activity.

The authors have addressed issues with clear statements about statistical analyses. Additionally, it is important to keep in mind that even small changes to dendrites have a great effect on neural circuits. Regardless, the changes observed are, in fact, not small (Figure 4).

I also have no issues with the use of rat and mouse neurons. The authors give ample evidence for conservation of SLAMR orthologs in a number of species. The authors are also correct in the neurons cannot be cultured from animals beyond a few days after birth. The neurons do not survive in culture. The authors have now clarified which neurons they use in each experiment.

Additionally, it is common that Western blot band detection can vary, and the patterns seen in Figure 6 are common.

Reviewer 2

The authors have incorporated discussion of Delorme et al, 2021 and others into their Discussion.

Although it is correct that adjusted p values should be reported, the authors report nominal p value but then do report adjusted p values in Figure 1C and Supplementary Figure S1, alleviating previous concerns.

The authors also referenced previous papers as support for their analysis of morphological changes to spines, which is appropriate.

All additional minor comments were addressed.

Reviewer 3

Most of the comments referred to the tempering of the authors' statements. The authors were very responsive and made edits in response to the reviewer. Additionally, the authors clarified methods and added additional experiments (i.e. evaluation of puromycin labeling in SLAMR-KD neurons) and gave adequate support for experimental design and speculation (i.e. no movement of particles in dendrites and potential SOX9 regulations).

In sum, the authors have done extensive work and given support to address the reviewers' comments.

REVIEWERS' COMMENTS

Reviewer #1 (Remarks to the Author):

The authors have gone to great lengths to address the questions and concerns raised as well as modify the main text. All of these responses make sense and help provide context to the results the authors are focusing on. I don't have any further questions.

Our Response: We thank this reviewer for their previous comments to help improve the manuscript.

Reviewer #3 (Remarks to the Author):

The authors have satisfactorily addressed all of my concerns from the prior round of review.

Our Response: We thank this reviewer for their previous comments and suggestions to help improve the manuscript.

Reviewer #4 (Remarks to the Author):

This revised manuscript reports exciting data to support a role for a novel lncRNA, SLAMR, in dendritogenesis, spine morphology, and translation. SLAMR associates with CaMKII α protein, demonstrating a mechanism for its action in neurons. The experiments range from primary neuronal culture to neurobehavioral studies in mice. The work is of interest to a broad audience.

The authors have added a large amount of new data in response to previous criticisms, which I detail below.

Reviewer 1

In summary, they have added data to support the following:

1. SLAMR modulates translation and dendritic morphology as evidenced by both overexpression and knockdown experiments.
2. Identification of CamKII alpha as a SLAMR protein interactor.
3. SLAMR is actively transported in dendrites and spines and that this transport is regulated by activity.

The authors have addressed issues with clear statements about statistical analyses. Additionally, it is important to keep in mind that even small changes to dendrites have a great effect on neural circuits. Regardless, the changes observed are, in fact, not small (Figure 4).

I also have no issues with the use of rat and mouse neurons. The authors give ample evidence for conservation of SLAMR orthologs in a number of species. The authors are also correct in the neurons cannot be cultured from animals beyond a few days after birth. The neurons do not survive in culture. The authors have now clarified which neurons they use in each experiment.

Additionally, it is common that Western blot band detection can vary, and the patterns seen in Figure 6 are common.

Reviewer 2

The authors have incorporated discussion of Delorme et al, 2021 and others into their Discussion.

Although it is correct that adjusted p values should be reported, the authors report nominal p value but then do report adjusted p values in Figure 1C and Supplementary Figure S1, alleviating previous concerns.

The authors also referenced previous papers as support for their analysis of morphological changes to spines, which is appropriate.

All additional minor comments were addressed.

Reviewer 3

Most of the comments referred to the tempering of the authors' statements. The authors were very responsive and made edits in response to the reviewer. Additionally, the authors clarified methods and added additional experiments (i.e. evaluation of puromycin labeling in SLAMR-KD neurons) and gave adequate support for experimental design and speculation (i.e. no movement of particles in dendrites and potential SOX9 regulations).

In sum, the authors have done extensive work and given support to address the reviewers' comments.

Our Response: We thank this reviewer for their enthusiasm for our work and for helping to verify that we have met the other reviewers' concerns.